

# Homogenizing and Estimating the Uncertainty in NOAA's Long Term Vertical Ozone Profile Records Measured with the Electrochemical Concentration Cell Ozonesonde

Chance W. Sterling[1,2], Bryan J. Johnson[1], Samuel J. Oltmans[1], Herman G. J. Smit[3], Allen F. Jordan[1,2],
Patrick D. Cullis[1,2], Emrys G. Hall[1,2], Anne M. Thompson[4], Jacquelyn C. Witte[4, 5]

[1]National Oceanic and Atmospheric Administration, Boulder, CO, 80305, United States
[2]Cooperative Institute for Research in Environmental Sciences, Boulder, CO, 80309, United States
[3]Forschungszentrum Jülich GmbH, Institute of Energy and Climate Research: Troposphere, Jülich, Germany
[4]NASA Goddard Space Flight Center, Greenbelt, MD, USA
5 SSAI, Lanham, MD, USA

*Correspondence to*: Chance W. Sterling (chance.sterling@noaa.gov)

**Abstract.** NOAA's program of long term monitoring of the vertical distribution of ozone with Electrochemical Concentration Cell (ECC) ozonesondes has undergone a number of changes over the 50 year record. In order to produce a homogenous data set, these changes must be documented and where necessary, appropriate corrections applied. This is the first comprehensive and consistent reprocessing of NOAA's ozonesonde data records that corrects for these changes using the rawest form of the data (cell current and pump temperature) in native resolution as well as a point by point uncertainty calculation that is unique to each sounding. The reprocessing is carried out uniformly at all eight ozonesonde sites in NOAA's network with differences in sensing solution and ozonesonde types accounted for in the same way at all sites. The corrections used to homogenize the NOAA ozonesonde data records greatly improve the ozonesonde measurements with an average one sigma uncertainty of ±4-6% in the stratosphere and ±5-20% in the troposphere. A comparison of the integrated column ozone from the ozonesonde profile with co-located Dobson spectrophotometers total column ozone measurements shows agreement within ±5% for >70% of the profiles. Very good agreement is also found in the stratosphere between ozonesonde profiles and profiles retrieved from the Solar Backscatter Ultraviolet Instruments (SBUV).

## 1 Introduction

Soon after the discovery of ozone in the atmosphere by Schönbein in 1840 (Bojkov, 1986) the first semi-quantitative measurements of ozone were made by exposing starch/iodide test papers to outdoor air using the Schönbein "ozonometer" developed in 1845 (Bojkov, 1986; Graedel, 1993) . The scientific interest over this new form of oxygen resulted in a broad range of studies that focused on the role ozone plays in the atmosphere and refining measurement techniques. Accurate measurements of ozone by wet-chemical methods using a bubbler and aqueous potassium iodide (KI) were developed. A. Levy, using a bubbler/titration technique, began daily surface ozone measurements at the Montsouris Observatory in France



(Volz and Kley, 1988) that continued for 34 years from 1876-1910. The wet-chemical method based on the fast reaction of ozone and iodide in a neutral buffered KI solution remained a standard measurement method up through the 1970s when ozone studies focused on air quality in cities along the California urban corridor. By the 1980s, ultraviolet photometry became the new standard for measuring surface ozone (Oltmans, 1981). However, the neutral potassium iodide method remained a useful

technique for balloon-borne vertical profile measurements of ozone. A number of balloon-borne techniques were tested and employed to measure the ozone vertical profile. Early ozonesondes included optical (Külke and Paetzold, 1957; Kobayashi et al., 1966), chemiluminescent (Regener 1964) and electrochemical (Brewer and Milford, 1960) sensors. Each of these methods exhibited limitations in terms of making an accurate quantitative measurement of the ozone profile as well as somewhat cumbersome preparation procedures (Moreland, 1960). The Electrochemical Concentration Cell (ECC) ozonesonde eventually

emerged as a widely used, relatively simple method to measure accurate ozone profiles from surface to 30-35 km above sea level when the sensing instrument is interfaced with a balloon-borne meteorological radiosonde (Komhyr et al., 1969, 1995).

**Importance of vertical profile measurements**

Ozonesondes have played an important role in monitoring the stratospheric ozone layer where harmful solar ultraviolet radiation is absorbed by ozone thus protecting the biosphere (Stolarski, 2001). Although ozonesonde sites around the globe

are relatively sparse and not uniformly distributed, selected long-term data sets have been compared and analyzed for trends. Ozone trend estimates at selected altitude intervals were first reported by Logan (1985,1994), Tiao et al. (1986), London and Liu (1992), and Oltmans et al. (1998) using data from several ozonesonde sites that had compiled 2 or more decades of data with 1-2 balloon flights per week. More recently, ozonesonde data have been used in developing ozone climatologies (Tilmes et al., 2012; Hassler et al., 2013; Sofieva et al., 2014) and validating satellite tropospheric retrievals (Verstraeten et al., 2013;

Martins et al., 2015; Thompson et al., 2012; Hubert et al., 2016). Ozonesonde data has been used for analyzing long range transport of tropospheric ozone (Cooper et al., 2011) and stratospheric/tropospheric exchange events (Terao et al., 2008; Langford et al., 2012). Ozonesondes have shown the characteristic view of the zero ozone depletion layers during the Antarctic ozone hole monitoring (Hofmann et al. 2009, Hassler et al., 2011) as well as revealing Arctic stratospheric ozone loss rates (Rex et al., 2002). An important question at this time is how stratospheric ozone responds to climate variability in the future

(Harris et. al, 2015).

Data homogenization is necessary for long-term ozone profile records that have gone through instrument and operating procedure changes in order to provide consistent data with reduced uncertainties and offsets. The framework for addressing global data quality and consistency from all ozone profile measurement techniques (Hassler et al., 2014) came from the SPARC/IO3C/IGACO-O3/NDACC (SI2N) initiative designed in 2011. (SPARC - Stratosphere-troposphere Processes And

their Role in Climate, IO3C - International Ozone Commission, IGACO-O3 - Integrated Global Atmospheric Chemistry Observations – Ozone, NDACC - Network for the Detection of Atmospheric Composition Change). In order to directly address the quality of ozonesonde data records, a subsection of the SI2N initiative presents the OzoneSonde Data Quality Assessment (O3S-DQA) and homogenization of the balloon-borne ozonesonde records. The O3S-DQA report by Smit and the O3S-DQA panel (2012) outlined the following goals: (a) produce a fully homogenized ozonesonde data set from selected long-term sites



by removing biases from known changes in instruments and applying transfer functions for sensor solution changes in operating procedures; (b) clearly document the process and address quality of the individual ozonesonde profiles; (c) reduce uncertainty from 10-20% to 5-10% and (d) include uncertainty in the reported ozonesonde flight data.

The recommended guidelines for homogenizing long-term ozone records and standardizing current operational procedures

(Smit and the O3S-DQA panel, 2012) are based on several ozonesonde intercomparison projects in the laboratory and field. The laboratory experiments were conducted at the World Calibration Centre for Ozonesondes (WCCOS) in Jülich, Germany. The WCCOS is an environmental chamber capable of simulating various ozone profiles with a UV ozone photometer reference measurement (Proffitt and McLaughlin, 1983). Jülich OzoneSonde Intercomparison Experiments (JOSIE) were conducted in 1996, 2000, and 2009. These experiments had slightly different set ups and goals, but usually focused on comparing different

ozonesonde models from the manufacturers (Science Pump Corporation (SPC) and EN-SCI Corp), different sensing solution recipes (Smit, 2007; Smit and Sträter, 2004a), and different Standard Operating Procedures (SOP's) (Smit and Sträter, 2004b). A field test intercomparison of ozonesonde models and sensor solutions was conducted during the World Meteorological Organization (WMO)-sponsored Balloon Experiment on Standards for OzoneSondes (BESOS) campaign held at the University of Wyoming Balloon Facility in Laramie, Wyoming, USA (Deshler et al., 2008). The BESOS balloon gondola

carried 12 ozonesondes (6 EN-SCI and 6 SPC) alongside the JOSIE UV ozone reference instrument.

These intercomparison projects showed that when the same sensing solution was used the EN-SCI Corp model ozonesondes measured approximately 5% higher ozone than SPC ozonesondes; when the same ozonesonde type was used the standard 1% KI buffered sensor solution measured approximately 5% higher ozone than the half percent solution (Smit et al., 2007; Deshler et al., 2008). Deshler et al. (2017) provides linear transfer functions to apply to ozonesonde data to account for changes in

ozonesonde model or sensor solutions based on the WMO and JOSIE intercomparisons and other multi-ozonesonde comparison flights done by individual ozonesonde groups.

Several ozonesonde sites have published results of homogenized records, including the Canadian ozonesonde (Tarasick et al., 2016) and the Southern Hemisphere ADditional OZonesondes (SHADOZ) networks (Thompson et al., 2017; Witte et al., 2017a). Homogenization of sites that switched from Brewer Mast type ozonesondes to ECC ozonesondes include Uccle

(Lemoine & De Backer, 2001) and Payerne Aerological Station (Stübi et al., 2008). Dual and multiple ozonesonde flight data at Sodanklyä were used to homogenize data and evaluate trends from different Arctic ozonesonde sites (Kivi et al., 2007).

Here we present the homogenization procedure and results for the NOAA/ESRL/GMD ECC ozonesonde network consisting of eight long-term monitoring sites. While this effort represents the first reprocessing of these data that attempts to account for all known contributors to inhomogeneity and biases in the data in a systematic way, several earlier versions of the data have

been available that tried to account for some of the inhomogeneity and biases. Previous versions of the ozonesonde data archived at NDACC, SHADOZ, and the WOUDC accounted for pump efficiency losses, impact of sensing solution composition, unrealistic background current measurements, and ozonesonde manufacturer differences. A better quantification of these factors as well as a number of others are discussed and their incorporation into the reprocessed data set is presented here.





## 1.1 Ozonesonde Instrument and Standard Operating Procedures Changes

Homogenizing long term records of ECC ozonesonde data begins with reviewing the upgrades and changes in instrument design and SOP's. Table 1 lists the different models and manufacturing dates of ozonesondes made by Science Pump Corporation and EN-SCI Corporation. There have been seven changes in the manufacturer model design. NOAA's first

ozonesondes in 1967 included the earliest version, 1A ozonesonde. Ozonesonde launches were infrequent until after 1985 when three sites began launching regular weekly ozonesonde flights, eventually using all of the ozonesonde models (1A, 3A, 4A, 5A, 6A, 1Z and 2Z) up through the present time. There were two major design changes in the ozonesonde models. One was the introduction of the more efficient, cylindrical cross-section pump in the 4A ozonesonde. Section 3.2 outlines the method for accounting for this change. The second was moving the position of the thermistor to more accurately measure the

true gas temperature flowing through the pump chamber. Section 3.3 outlines the method of applying a correction algorithm for adjusting box temperature to gas temperature in the pump.

The guidelines for preparing an ozonesonde for flight include the manufacturers instruction manuals and the WMO SOP's (Smit and ASOPOS panel, 2014), which are based on workshop reviews of JOSIE and BESOS ozonesonde testing. During the long- term NOAA record there have been adjustments to the guidelines or SOP's that include, for example, how to measure

the cell background and changes in the radiosonde interfaced with the ozonesonde. By far, the two changes that have the greatest impact on ozonesonde measurement accuracy are the pump flow rate efficiency correction curve applied and adjustments to composition of the sensor solutions (Johnson et al., 2002; Smit et al., 2007).

## 1.2 Sensing Solution Type Changes

Changes in the ozonesonde sensing solution compositions (Table 2) used are a significant factor that needs to be taken into

account since this affects the chemistry of the ozone iodide reaction stoichiometry. The sensor solution composition recipes used by the early ECC ozonesondes originated from the wet chemical, iodometric techniques (Bartel and Temple (1952); Littman and Benoliel (1953); Saltzman and Gilbert (1959); Boyd et al. (1970)). The method involves the absorption of ozone and oxidation of iodide ions to iodine ($I_2$) with an overall stoichiometric balance of 1:1 ($O_3 = I_2$) as shown in Eqn. (1):

2    $KI + O_3 + H_2O \rightarrow 2\,KOH + I_2 + O_2$ (1)

25   The iodine product can be measured by titration, colorimetric methods, or coulometry. For example, Saltzman and Gilbert (1959) used kinetic colorimetric detection of iodine when testing various absorption reagents. They found the 1% KI with neutral phosphate buffering gave the best result, close to the ideal 1:1 stoichiometry. However, they noted that additional iodine was produced, referred to as slow color product, and theorized that the side reaction sequences are complex and may involve the buffers producing excess iodine through slower, secondary reactions. High ozone readings were also reported in a U.S.

30   Environmental Protection Agency (EPA) workshop which was held to resolve some of the iodometric measurement issues for surface ozone monitoring networks. The workshop participants included the EPA, National Bureau of Standards, NASA, several California air quality departments and research organizations. A representative from NOAA, and developer of the ECC ozonesonde, Komhyr (1969), presented the ECC ozonesonde method. The workshop Summary Report (Clements, 1975)





focused on the high ozone readings related to sensing solution composition, pH and type of bubbler used. The Los Angeles County Air Pollution Control District (Clements, 1975) reported that a 2% KI unbuffered reagent, used to calibrate their standard ozone measurement procedure, gave optimum results due to the apparent suppression of artifact high ozone. The earliest laboratory testing of the 2% KI unbuffered aqueous KI solution by Birdsall, et al. (1952) showed precise results when

measuring very high ozone concentrations. The 2% KI unbuffered sensor solution was tested for use with ECC ozonesondes (Johnson, et al., 2002) showing similar results comparing surface ambient ozone monitoring comparison with a standard UV monitor and dual ozonesondes. The 2% KI, unbuffered solution appeared to eliminate the overestimation of ozone due to the slow side reaction increasing the stoichiometry. However, the 2% KI, unbuffered solution typically was lower in total column ozone when compared to Dobson spectrophotometer measurements and exhibited a higher occurrence of cell sensor spikes.

The decision was made to return to a modified version of the original neutral 1% KI with 2.5% KBr solution, but the buffering agent was reduced to very low concentrations to lessen the secondary reaction, yet maintain a constant pH of 6.8 (Johnson et al., 2002). In the past various solution recipes have been linked with the choice of the correction factor for pump efficiency loss at higher altitudes (lower pressures) (Komhyr, 1986; Komhyr et al., 1995). This non-physical linkage is abandoned in this effort. A full discussion of correction for the pump efficiency loss is discussed in a later section.

**1.3  Data Acquisition and Radiosonde Changes**

The ECC ozonesonde is interfaced with a radiosonde to transmit the ozone data to the surface and have an accurate measurement of atmospheric conditions, most importantly ambient pressure, temperature and relative humidity. VIZ radiosondes were used during the analog era (1967-1991) and gave a data resolution of approximately one minute or 300 meters. The VIZ radiosondes used a hypsometer for pressure measurements at altitudes above ~20 hPa at Boulder, Hilo, and

South Pole from 1986-1989. Pressure measurements with the accompanying hypsometer were accurate to ±0.2 hPa. (Conover and Stroud, 1958). VIZ radiosondes were tested in a number of radiosonde intercomparison campaigns with average pressure errors falling in the range of 1-3 hPa at pressures less than 15 hPa (Schmidlin et al., 1982).

The RS-80 radiosonde manufactured by Vaisala was used by NOAA from 1991 until 2009 when transition to the iMet-1 radiosondes began. The RS-80's allowed for digital data acquisition when paired with an electronics board attached to the

ozonesonde. Initially the TMAX electronics board was used to couple the ozonesonde to the RS-80 radiosonde and was capable of measuring and transmitting data every seven seconds. The V2 electronics board introduced in 1998 improved the electronic components and increased the time resolution to one second data. The current radiosonde being used by NOAA is the iMet-1 manufactured by International Met Systems. i-Met radiosondes are equipped with a GPS receiver. Comparing the geometric altitude of the GPS to the geometric altitude calculated from the pressure, temperature, and relative humidity from the

radiosonde allows for an accurate pressure offset to be applied to the pressure sensor. The geometric altitude is only used for correcting the pressure sensor; the geopotential altitude is reported in all data files. A majority of flights conducted using RS-80 radiosondes did not have a GPS receiver attached. The RS-80 pressure sensors are known to have degraded over time. Several techniques were employed to evaluate possible errors in the pressure reading and make corrections. Radiosonde



pressure readings at the surface were compared with an accurate surface pressure measurement. Testing of a number of RS-80 radiosondes in an altitude chamber showed that the pressure offset at 7 mb was on average 75% of the pressure offset observed at the surface. This method of determining the pressure offset was used for all RS-80 radiosondes from 2008 – 2011 (approximately 1200 profiles in the NOAA long term network). Before 2008, the RS-80's pressure sensors were newer and

more accurate. The SkySonde processing software allows for comparing temperature profiles from nearby meteorological soundings to the temperature profile measured by the RS-80. RS-80's with large pressure offsets (>2 mb) could be identified and corrected using this comparison. A more thorough investigation into the non-GPS radiosondes pressure offsets could improve the accuracy and variation of the ozone profile measurement, especially the upper portion of the profile. Additionally, radiosonde pressure uncertainties were not included in the overall uncertainty calculations. That is beyond the scope of this

analysis.

## 2 Procedures and Calculations

### 2.1 Approach

NOAA followed the WMO reprocessing recommendations and guidelines when applicable (Smit and the O3S-DQA panel, 2012). However, NOAA uses a unique sensor solution recipe and measured its own pump efficiencies which necessitated

deriving corrections for these unique cases for the NOAA and many of the SHADOZ ozonesonde data records (Thompson et al., 2012; 2017). The ozonesonde equation for calculating the ozone partial pressure is determined by Faraday's first law of electrolysis and the ideal gas law shown in Eqn. 2:

$$P_{O_3} = \frac{R}{2*F} * [I_M - I_{BG}] * \frac{1}{\Phi_P} * T_P * \frac{1}{\eta_{OS}}$$    (2)

The first term is an empirical constant where R is the universal gas constant and F is the Faraday's constant. The two in the

denominator represents the two electrons being delivered to the electrical circuit of the sensing cell for every ozone molecule reacted, assuming a 1:1 stoichiometry. The remaining variables in Eqn. 2 are the measured cell current ($I_M$) and the background cell current ($I_{BG}$) in microamps, the pump flowrate ($\Phi_P$) in cm³/s, the pump temperature ($T_P$) in degrees Kelvin, and the ozone sensor efficiency ($\eta_{OS}$). The cell currents, the pump flowrate and the pump temperature can be measured directly and independently. The ozone sensor efficiency ($\eta_{OS}$) is a measure of how efficiently gaseous ozone molecules bubbled through

the ozone sensor are converted to electrons and cannot me measured directly. Instead it is measured by comparison to the reference ozone photometer at the WCCOS.

In order to homogenize the NOAA ozonesonde data record and account for changes in ozonesonde types and sensing solutions, a two-step approach was taken. First, the variables that can be quantified directly were treated consistently through the entire record. Individual ozonesonde data profiles were quality controlled by correcting or flagging erroneous measurements in the

measured cell current, pump temperature and radiosonde pressure. Failed ozonesonde flights were screened out or data was



cut off at altitudes where the ancillary data such as battery voltage or pump temperature indicated a failure. Profile altitude errors from radiosonde pressure offsets (before GPS geometric altitude became available) were fixed by applying corrections to the pressure sensors as noted earlier. Erroneously measured variables such as cell current backgrounds were fixed systematically, changes in how variables are measured such as pump temperature were accounted for, and climatological or

average values were assumed in instances where a variable was not used in historic data such as for pump flowrate corrections. Second, the ozone sensor efficiency was determined for the different sensing solution and ozonesonde types from the comparisons of the ozonesonde and the reference UV photometer at JOSIE. The ozone sensor efficiencies were then applied appropriately to all data files to create a consistently calculated, homogenous data set. This approach homogenizes the data to the ozone photometer at WCCOS for each solution type and ozonesonde type by applying a unique ozone sensor efficiency.

This is in contrast to the approach of homogenizing the record to an ozonesonde type/solution type/pump efficiency pairing found to have good agreement with the reference photometer.

Figure 1 shows the many changes to the NOAA ozonesonde record. The changes in solution, ozonesonde type, digital to analog data acquisition, and an observed change in the cell current backgrounds led to a logical division of NOAA's ozonesonde data record into five eras.

Era 1 is the earliest portion of the analog era from 1/1967 to 6/1/1982 that primarily used 1A and 3A ozonesonde types and the 1.0% KI, 1.0x Buffer solution but also 1.5% 1.5x Buffer Solution. The change from 1.5% 1.5x Buffer Solution to 1.0% KI, 1.0x Buffer Solution was not well documented on individual flight records but the soundings after 1972 used the 1.0% KI, 1.0x Buffer solution exclusively. These earliest data are treated similarly to the second era based on lack of information that would improve the corrections. This is accounted for with an increase in the uncertainty. Era 2 is the period from 6/1/1982 to

1/1/1991 that used 1.0% KI, 1.0x Buffer Solution and primarily 4A ozonesondes. Era 3 is the period from 1/1/1991 to 1/1/1998 that used 1.0% KI, 1.0x Buffer Solution and primarily 5A, 6A and 1Z ozonesondes. Era 3 was also the beginning of digital data acquisition for NOAA. Era 4 is the period between 1/1/1998 and 6/1/2005 that used 2% KI, No Buffer Solution and was divided into two sub eras. Era 4a used Z ozonesondes and Era 4b used 6A ozonesondes. This era was subdivided due to observed ozonesonde type bias between 6A and Z ozonesondes (Deshler et al., 2008; Smit and Sträter, 2004b). Era 5 is the

current era starting 6/1/2005 that uses the 1.0% KI, 0.1x Buffer Solution and primarily 2Z ozonesondes.

The historic JOSIE data sets were valid in quantifying the ozone sensor efficiency for these different eras because the ozonesonde measurements taken at JOSIE were consistent with the ozonesondes, solutions, and standard operating procedures being used by NOAA at the time.

## 2.2 Meta Data and File Types

Before homogenizing the NOAA ozonesonde network all of the necessary metadata that was available was collected and added to the digital data files, all data files were converted to a common, editable file type which includes the rawest form of the data (cell current and pump temperature) in all files. This was a time and labor intensive process that required the development of a new data acquisition and processing software called SkySonde. The one minute analog data was read from chart records and





digitized. It was common to only calculate significant and designated levels in the analog chart record data. However, NOAA digitized every one minute data point for all 1,179 analog data files in the NOAA ozonesonde record. In the analog data, the commutator was powered by the pump motor. Changes in the motor speed resulted in changes in the time resolution of the data. With careful consideration, the changing motor speed was accounted for by multiplying the cell current by a motor speed

correction factor.

Once all data files were in a common format and included the rawest form of the measurement, corrections could be applied in batch. This first step was a major achievement and paved the way for quickly and easily making changes to the entire data set. This will also allow for future reprocessing of the data if additional information on the characteristics of the ozonesondes (and perhaps radiosondes) performance are obtained.

**2.3 Reverse-calculating Cell Current**

Early on when a TMax interface board was used, the data acquisition software did not output the cell current. In order to include cell current in the data files, a reverse calculation of cell current was performed. Careful consideration is required to back calculate cell current correctly. All of the necessary variables needed to back calculate the cell current from the ozone partial pressure were available in the data file. Thus, this calculation was carried out with negligible error.

**3 Variables for Calculating Ozone Partial Pressure**

**3.1 Measured Cell Current and Background Cell Current**

The measured cell current is the electrical current that is produced by the ozone sensor cell and measured by the electronics board throughout the flight. The time resolution and acquisition systems have changed over the record, but the variable has not. The background cell current is the residual current produced by the ozonesonde when ozone free air is sampled and is

determined during the flight preparation. A detailed analysis of the source of ozonesonde background current revealed that it was not oxygen dependent. (Thornton and Niazy, 1982, 1983) Vömel (2010) demonstrated that cell current background declines for up to 90 minutes when ozone free air is sampled after exposure to ozone; as well as, the importance of the background in the very low ozone observed in the tropics. It is theorized that this long decaying background is related to the slow side reactions of the phosphate buffer.

Current recommended SOP's call for three cell current background measurements to be recorded. Ib0 is recorded after the ozonesonde has been sampling ozone free air for 10 minutes before ozone exposure, Ib1 is recorded after sampling ozone free air for 10 minutes after ozone exposure, and Ib2 is recorded directly before launch with the goal of achieving a low and constant reading (Smit and the ASOPOS panel, 2014). Historically, NOAA has always used Ib2 for calculating ozone. For portions of the early record, Ib2 was the only cell current recorded. References to the cell current backgrounds in this work are to Ib2.

NOAA's SOP's were to use an ozone destruct filter at the launch site to establish the background current of the cell. These filters degraded over time, especially in humid marine environments, causing many erroneous background measurements.



When an ozone free air source is used, the background is dependent primarily on the solution type and also on the ozonesonde type. Additionally, cell current backgrounds decreased substantially around 1991 (Smit and the O3S-DQA-Panel, 2012). These facts align well with the eras since they are primarily based on solution type changes. The drop in backgrounds in 1991 led to grouping Era 1 and Era 2 together and leaving Eras 3, 4, and 5 separate for the cell current background analysis as seen in

Figure 2. To correct the erroneously high background measurements, a background reduction system was created based on an average cell current background and standard deviation for each era. If the measured cell current background was greater than the average background plus one standard deviation, the background measurement was replaced by the average value.

The three longest running stations (Boulder, South Pole, and Hilo) have had the most consistent and highest quality ozone preparation and documentation. Figure 2 shows the histograms of the originally measured backgrounds after exposure to ozone,

Ib2, at these three sites. When these histograms are compared to the backgrounds taken at intercomparisons, it is clear that Era 1/2 and Era 3 were measuring a large number of erroneous backgrounds. The statistics on the backgrounds in these eras (Panel A and B of Figure 2) are not indicative of the actual backgrounds and thus are not used for the background reduction. Instead, the mean and standard deviation found at intercomparisons where high quality background measurements were taken are used. For Era 1 and 2, the mean background was taken as 0.09 ±0.02; for era 3, the mean background was taken as 0.05 ±0.02 (Smit

and the O3S-DQA-Panel, 2012). The backgrounds during Era 4 and Era 5 were measured more carefully and the results aligned with the findings in the JOSIE and BESOS campaigns (Smit et al., 2007; Deshler et al., 2008). The background data and statistics from the three stations found in Panel C and D of Figure 2 were used to determine the upper limits for the coinciding era. This resulted in retaining a higher percentage of the originally measured backgrounds.

**3.2 Ozonesonde Pump Flowrate**

All ozonesonde pump flowrates were measured with a 100 ml bubble flow meter at the station by averaging five stopwatch measurements. The measured flow rate must be corrected for two issues. A correction must be applied to account for the humidification of air being measured and the cooling of the air from the pump temperature to the temperature of the air being measured in the bubble flow meter (Smit and ASOPOS panel, 2014). Second, a correction must be applied to the volumetric pump flowrate to account for the loss of efficiency of the ozonesonde pump at pressures below 300 hPa. The volumetric pump

flow rate is calculated from Eqn. 3:

$$\Phi_P = \Phi_{P,Meas} * C_{PF,SM} * \eta_{PF,LP} \tag{3}$$

$\Phi_{P,Meas}$ is the volumetric pump flowrate measured at the surface in cm$^3$/s, $C_{PF,SM}$ is the pump flowrate correction for the surface measurement, and $\eta_{PF,LP}$ is the pump flowrate efficiency at low pressures. Historically, the pump flowrate efficiency has been reported as pump correction factors (PCF's) which is the inverse of the pump flowrate efficiency.





### 3.2.1 Correction for Surface Measurement of Pump Flowrate

The pump flowrate correction for the surface measurement, $C_{PF,SM}$, is calculated by Eqn. 4:

$$C_{PF,SM} = 1 - C_{P,H} + C_{P,TD} \tag{4}$$

$C_{P,H}$, the correction for the humidification effect, is subtracted from 1 because the flowrate needs to be reduced to account for

the added water vapor. $C_{P,TD}$, the correction for the temperature difference in the pump and the air being measured, is added because the volume of air has been reduced due to the cooling from the pump to the bubble flow meter.

During the flowrate measurement, the ozonesonde samples the filtered air exiting the test unit. The volume of air being measured becomes saturated with water vapor as it is bubbled through the sensor solution and travels along the wetted walls of the bubble flow meter. The humidification effect is calculated by Eqn. 5:

$$C_{P,H} = \left[ 1 - \frac{RH_{TU}}{100} \right] * \frac{P_{H_2O,Sat}(T_{FM})}{P_{FM}} \tag{5}$$

The volume of water vapor added to the air being measured is dependent on the ambient pressure in the flow meter, the vapor pressure of the air in the flow meter, and the relative humidity of the air entering the ozonesonde pump which is assumed to be the relative humidity of the air exiting the test unit, $RH_{TU}$. In climatological cases when ozone destruct filters were used, $RH_{TU}$ is assumed to be the ambient relative humidity. The saturated water vapor pressure at the temperature of the air in the

flow meter, $P_{H_2O,Sat}(T_{FM})$, is calculated using the Hyland and Wexler approximation (Hyland and Wexler, 1983). The air temperature in the flow meter, $T_{FM}$, is assumed to be the ambient temperature.

The correction for the pump temperature and air temperature in flow meter difference is assumed to be adiabatic compression and is approximated by Eqn. 6:

$$C_{P,TD} = \frac{T_P - T_{FM}}{T_{FM}} \tag{6}$$

The pump temperature during the flowrate measurements has only been recorded since July 2016 making individually calculated flowrate corrections impossible for a large portion of the NOAA record. A climatological value is used instead. NOAA introduced the flowrate correction between 1998 and 2000 depending on the station. This is when metadata on the laboratory conditions began to be kept. The data were often not accurate, or incomplete, or not recorded at all. A monthly climatology was calculated based on the lab conditions for each site. The data were screened for quality and notable changes

were accounted for. The pump temperature difference compared to the ambient temperature is assumed to be three degrees K for the climatology. Knowing the monthly average ambient pressure, temperature and relative humidity and the pump to ambient temperature difference, the climatological flowrate correction can be calculated for each site.

Since 2010, NOAA has used a Drierite air purifier/desiccant filter rather than canister ozone destruct filters to produce a zero ozone air source at the Boulder, Trinidad Head, and Fiji sites. The desiccant strips the air of all water vapor. With a stable lab

temperature and pressure and zero humidity air being sampled for the flowrate measurement, the flowrate correction becomes nearly constant.





Figure 3 shows the different climatological flowrate corrections. The flowrate corrections in Fiji range from approximately 100.1% to 100.5% and for South Pole from 97.5% to 97.9%. The sites' seasonal variation is low and may be trivial; however, the site to site difference can be greater than 3% making the flow rate correction for the surface measurement necessary for a uniform homogenization of all sites.

### 3.2.2 Volumetric Pump Flowrate Efficiency Loss at Low Pressures

As the ambient pressure decreases during flight, the efficiency of the ozonesonde pump begins to decline due to leakage, the dead volume in the piston, and the back pressure exerted on the pump by the sensor solution (Komhyr and Harris, 1971, Steinbrecht et al., 1998, Johnson 2002). Smit and the ASOPOS panel (2014) recommends using the Komhyr (1986) or Komhyr et al. (1995) pump efficiency corrections. This recommendation was based on the observed agreement of the ozonesonde to the reference ozone photometer at the JOSIE and BESOS intercomparison campaigns when the Komhyr (1986) and Komhyr et al. (1995) pump efficiencies were paired with a 1% KI, 1.0x Buffer Solution or 0.5% KI, 0.5x Buffer Solution respectively (Smit et al., 2007; Deshler et al., 2008). The good agreement observed using the smaller pump efficiencies reported by Komhyr (1986) and Komhyr et al. (1995) compared to the NOAA pump efficiencies (Johnson et al., 2002) is attributed to the compensating effect of the positive bias in the ozone sensor efficiency created by the side reactions of the phosphate buffers used in all solutions, except 2% KI, No Buffer (Johnson et al. 2002). Here we treat the influence of the buffer separately and use the Johnson et al. (2002) pump efficiency measurements. Correction for the influence of the buffer is attributed to the ozone sensor efficiency and is covered later in this work.

The Komhyr (1986) pump efficiencies were measured with a similar apparatus as Torres (1981) with the assumption that the hydrostatic back pressure from the sensing solution and the pump dead volume were responsible for the loss of efficiency of the ECC pump. The Torres (1981) apparatus used the ozonesonde pump to pressurize a chamber to the expected hydrostatic back pressure at varying pressure levels. The Komhyr (1995) pump efficiencies assumed that the pump efficiency of an ozonesonde pump was 100% at all pressures if no back pressure was applied to it. The apparatus used to measure the Komhyr (1995) pump efficiencies used two competing ozonesonde pumps (one pumping into a sensing cell with 3 cm$^3$ of sensor solution and one without solution that had a variable speed motor). The motor speed was adjusted to equalize the flow rates to calculate the pump efficiency. Johnson et al. 2002 used an oil bubble flow meter to measure the unimpeded pump volumetric flow directly at low pressures to determine pump efficiencies. The University of Wyoming and the Japan Meteorological Agency accomplished this using a bag inflation method. Interestingly, very early on Komhyr and Harris (1971) measured the pump efficiency of 3A ozonesondes with a bag inflation method and determined the average 3A pump efficiency correction to be approximately 1.13 and 1.225 at 10 hPa and 5 hPa respectively. These PCF's agree nicely with the Johnson et al. (2002) PCF's of 1.145 and 1.260 at 10 hPa and 5 hPa respectively.

This agreement led to using the Johnson et al. (2002) "all average" for the 1A and 3A pump efficiencies in this work. The PCF averages for 5A, 1Z, and 2Z ozonesonde types were all within one standard deviation up to 10 hPa. The 6A average fell outside of one standard deviation. Due to this fact, 6A ozonesondes were processed with the Johnson et al. (2002) 6A average PCF's.



All other ozonesonde types are processed with the "all" average PCF's. An updated and more detailed study of the ozonesonde pump efficiency could provide reduced uncertainty in the pump flowrate and improved confidence in the consistency of the pump performance over time.

### 3.3 Ozonesonde Pump Temperature

An accurate measurement of the pump temperature is required to calculate the volume of air passing through the ECC pump. The location of the pump temperature measurement has changed multiple times. In the NOAA ozonesonde record, there are three possible configurations. For 1A, 3A, and 4A ozonesonde types, a rod thermistor at the base of the ozonesonde body was used. For the 5A ozonesonde type, a thermistor was epoxied to the surface of the pump block. For 6A, 1Z, and 2Z ozonesonde types, the thermistor was mounted inside a hole drilled in the pump block. In order to account for these changes, the WCCOS

conducted experiments comparing old pump measurement configurations to the new configuration and the new configuration to the internal piston temperature (Smit and the O3S-DQA-Panel, 2012). The pump temperature is calculated by adding the differences between configurations and inside of the pump block and the difference between the inside of the pump block and the internal piston temperature by Eqn. 7:

$$T_P = T_{P,Meas} + \Delta T_{P,CIB} + \Delta T_{P,CIP} \qquad (7)$$

For 1A, 3A, and 4A ozonesondes, the correction for the difference in the temperature measured by the rod thermistor at bottom of the ozonesonde and the temperature inside the pump block, $\Delta T_{P,CIB}$, is estimated by Eqns. 8, 9, and 10 (Komhyr & Harris, 1971 and Smit and the O3S-DQA-Panel, 2012):

$$\Delta T_{P,CIB} = [7.3 - 0.393 \, Log_{10}(P_{Air})] \qquad \text{at } P_{Air} \geq 40 \; hPa \qquad (8)$$
$$\Delta T_{P,CIB} = [2.7 + 2.6 \, Log_{10}(P_{Air})] \qquad \text{at } 6 < P_{Air} < 40 \; hPa \qquad (9)$$
$$\Delta T_{P,CIB} = 4.5 \qquad \text{at } P_{Air} \leq 6 \; hPa \qquad (10)$$

This set of transfer functions increases the pump temperature by 4.5-7 degree K. For the 5A ozonesonde, the correction for the difference in the temperature measured by the thermistor epoxied to the pump base and the temperature inside the pump block is estimated by Eqns. 11 and 12 (Smit and the O3S-DQA-Panel, 2012):

$$\Delta T_{P,CIB} = [6.4 - 2.14 \, Log_{10}(P_{Air})] \qquad \text{at } P_{Air} > 40 \; hPa \qquad (11)$$
$$\Delta T_{P,CIB} = 3.0 \qquad \text{at } 3 \leq P_{Air} \leq 40 \; hPa \qquad (12)$$

For 6A, 1Z, and 2Z ozonesonde types the temperature measured is the temperature inside the pump block. For all other ozonesonde types, the measured temperature was corrected to the temperature inside the pump block by Eqns. 8-12. To obtain



the truest pump temperature, the difference in the temperature inside the pump block and the internal piston temperature, $\Delta T_{P,CIP}$, is estimated by Eqn. 13 (Smit and the O3S-DQA-Panel, 2012):

$$\Delta T_{P,CIP} = [3.90 - 0.80 \, Log_{10}(P_{Air})] \qquad \text{at } P_{Air} > 3 \, hPa \qquad (13)$$

After these pump temperature corrections are applied, the pump temperature used in Eqn. 2 for all ozonesonde types has
been transferred to the internal piston temperature, making the pump temperature measurements homogenous.

### 3.4 Ozone Sensor Efficiency

The ozone sensor efficiency, $\eta_{OS}$, is a measure of how efficiently gaseous ozone molecules are converted to electrons in the ozone sensor. The ozone sensor efficiency has two components, the absorption efficiency, $\eta_A$, and the conversion efficiency, $\eta_C$, and is calculated by Eqn. 14:

$$\eta_{OS} = \eta_A * \eta_C \qquad\qquad\qquad (14)$$

These variables are difficult to measure directly and independently, so they are measured and accounted for by comparing to an ozone photometer. The past JOSIE experiments are of great value in quantifying the ozone sensor efficiency for the different eras and ozonesonde type/sensing solution configurations. In order to accurately measure the ozone sensor efficiency by this comparison, the previously discussed variables used to calculate ozone partial pressure that can be quantified directly must be
treated identically in ozonesonde data record and the JOSIE comparison. For example, the pump flowrate efficiency used to calculate the partial pressure of ozone for the JOSIE experiments must be the same efficiencies used in the data record. Otherwise, the comparison and derived ozone sensor efficiency will be invalid. The differences seen in the ozonesonde and the ozone photometer at the JOSIE campaigns cannot be attributed to just one of these efficiencies. Therefore, the derived ozone sensor efficiency is accounting for both the absorption and conversion efficiency. The ozone sensor efficiency is
believed to be dominated by the stoichiometry of the reaction, but also the ozonesonde type. Therefore, deriving the ozone sensor efficiency for each era is the logical approach.

### 3.4.1 Absorption Efficiency

The absorption efficiency is a measure of how much of the gaseous ozone in the air pumped into the sensing solution is absorbed in the liquid phase. Davies et al. (2003) showed that when 3.0 cc of cathode sensing solution is used 100% of the
ozone pumped into the sensing solution is absorbed into the liquid solution. NOAA has exclusively used 3.0 cc of cathode sensing solution in its data record. Therefore, it is assumed that the absorption efficiency is one for the entire NOAA record. It should be noted that even if it is not one it is being accounted for by the ozonesonde/ozone photometer comparison and the derived ozone sensor efficiency correction factor.





### 3.4.2 Conversion Efficiency

The conversion efficiency is a measure of how much of the ozone molecules that are dissolved into the cathode solution are converted into electrons. A conversion efficiency of one would follow the stoichiometry of Eqn. 1 where one ozone molecule is converted into two electrons. Different sensing solutions and ozonesonde types result in different conversion efficiencies; the positive bias from the phosphate buffers is believed to cause the largest deviations in the conversion efficiency. There may be other unknown processes besides the stoichiometry that effect the conversion efficiency. These efficiencies are accounted for by measuring the ozone sensor efficiency.

### 3.4.3 Ozone Sensor Efficiency Correction Factors for each Era

The ozone sensor efficiency for Eras 1, 2, and 3 are treated the same as they all used 1.0% KI, 1.0x Buffer Solution, except for a few of the earliest flights in Era 1 that used the 1.5% KI, 1.5x Buffer Solution. As the amount of phosphate buffers used in the ozonesonde sensing solution increases due to evaporation, the positive bias in measured ozone values increases when compared to a photometer. As discussed earlier it is theorized that this is caused by a slower, secondary reaction pathway involving the phosphate buffers that increases the stoichiometry efficiency greater than 1 (Saltzman and Gilbert, 1959). When the measured Johnson et al. (2002) pump efficiencies are used with the 1% KI, 1.0x buffer solution, the calculated partial pressure of ozone has a positive bias greater than 15% above 20 km. Panel A of Figure 4 shows the positive bias measured on six simulated flights during JOSIE 1996.

This bias in the ozone sensor efficiency is assumed to primarily be due to the secondary reaction involving the buffer and is dependent on the amount of cumulative ozone exposure seen by the ozonesonde up to a given pressure (or altitude) level. The ozone sensor efficiency was estimated using the total accumulated column ozone as a measure of the exposure and is represented by Eqn. 15:

$$\eta_{OS} = (A + B * Total\ Column\ Ozone) \tag{15}$$

The Total Column Ozone is in units of atm*cm. It was determined that A = 1.02 and B = 0.4 produced the best agreement with the reference photometer. Panel B of Figure 4 shows the comparison with the reference ozone photometer after Eqn. 15 was applied to the six comparisons; the 2 km averages shows good agreement. The ozone sensor efficiency was then applied to all flights in which 1% full buffer solution was used. The ozone sensor efficiency for the 1% full buffer solution (Eqn. 15) is assumed to be a reasonable approximation for the 1.5% KI solution recipe that was used in flights prior to 1979.

Era 4 was subdivided because Deshler et al. (2008, 2017) and JOSIE 2000 (Smit and Sträter, 2004b) showed an ozonesonde type bias between 6A SPC ozonesondes and Z En-Sci ozonesondes when all other variables were constant. Era 4a used the 2% KI, No Buffer Solution unique to NOAA and SHADOZ with En-Sci Z ozonesondes. This sensing solution/ozonesonde type pairing exhibits a negative bias in ozone when compared to a UV photometer (Smit and Sträter, 2004a). It is believed that the lack of potassium bromide (KBr) and a buffering agent in the solution recipe cause this bias. Panel A of Figure 5 shows




the comparison of 3 ozone profile simulations at JOSIE 2000 for this ozonesonde/solution configuration. The ozone sensor efficiency for 2% KI, no buffer Solution with En-Sci Z ozonesondes is 0.98 throughout the entire profile. Era 4b also used the 2% KI, No Buffer Solution, but with SPC 6A ozonesondes. 6A SPC ozonesondes have been shown to measure 4% less than EN-SCI ozonesondes up to 30 hPa increasing to 10.3% at 10 hPa (Deshler et al., 2017). Deshler et al. (2017) did not account

for a difference in pump efficiencies for the 6A and Z ozonesondes. The pressure dependence of the bias is partially accounted for by the difference in the Johnson 2002 6A average and the Johnson 2002 all average pump correction factors used in this work. For Era 4b, the ozone sensor efficiency was estimated to be 0.94 through the entire profile as seen in Panel B of Figure 5. The difference between the 2Z and 6A ozonesondes observed by Deshler et al. (2008), JOSIE 2000 (Smit and Sträter, 2004b) and in Figure 5 have led NOAA to apply an ozone sensor efficiency of 0.96 to all 6A ozonesondes in addition to any needed

ozone sensor efficiency for a buffered solution.

Era 5 uses the 1.0% KI, 0.1x Buffer Solution and has yet to be compared to the standard ozone photometer at the WCCOS. Due to this lack of information, the ozone sensor efficiency for Era 5 is assumed to be one. Future work at JOSIE 2017 will provide the needed comparison data to quantify the ozone sensor efficiency for this era. Preliminary testing has shown that the reduced buffer amount in the 1.0% KI, 0.1x Buffer Solution has greatly reduced the positive bias exhibited by higher buffered

solutions.

## 4 Uncertainty of Ozone Partial Pressure

One of the primary objectives of the ozone data homogenization project was to estimate and calculate the uncertainty of the ozonesonde measurement. The partial pressure of ozone is a function of the measured cell current ($I_M$), the background current ($I_{BG}$), the volumetric flow rate of the pump ($\Phi_P$), the temperature of the pump ($T_P$), and the ozone sensor efficiency ($\eta_{OS}$). It is

assumed that the uncertainty in the calculation of the ozone partial pressure will be a composite of the individual uncertainties associated with each of the different variables. Because all systematic bias effects have been removed through this homogenization, it can be assumed that the uncertainties will be random and follow a random normal distribution. The uncertainty calculation must also account for the increased uncertainty incurred by homogenizing the data record and are included here. The overall relative uncertainty of $P_{O_3}$ is represented by the Gaussian law of error propagation in Eqn. 16 (Smit

and O3S-DQA Panel):

$$\frac{\Delta P_{O_3}}{P_{O_3}} = \sqrt{\frac{(\Delta I_M)^2 - (\Delta I_{BG})^2}{(I_M - I_{BG})^2} + \left(\frac{\Delta \Phi_P}{\Phi_P}\right)^2 + \left(\frac{\Delta T_P}{T_P}\right)^2 + \left(\frac{\Delta \eta_{OS}}{\eta_{OS}}\right)^2} \qquad (16)$$

A robust and accurate estimation of the ozone partial pressure uncertainty will be particularly beneficial when conducting trend analyses on this data set.





### 4.1 Uncertainty of Measured and Background Cell Current

The uncertainty in the measured cell current is a function of the errors and uncertainty of the electronics used for the measurement of the measured cell current. To estimate the uncertainty for the different digital interface boards, a reference current ranging from 0.025 µA to 7.5 µA was provided to the various interface boards and the measured cell current was recorded with the appropriate data acquisition system. The absolute value of the difference between reference current and the measured cell current was averaged for each reference current level. The average differences are nearly linear when the cell current is less than 1 µA and then increase proportionally to cell current when the cell current is greater than 1 µA. This characteristic makes the uncertainty best estimated by using a piecewise function. The estimated uncertainties associated with each interface board are summarized in Table 3. For the analog era, the measured cell current uncertainty also includes the uncertainty in the transfer of the measurement from the chart record to a digital file. The uncertainty in the measured cell current of the analog era was taken as 3% of the measured cell current when >1 µA and 0.03 µA for cell currents <1 µA (Komhyr and Harris, 1971).

If the cell current background was not reduced and remained the measured background, the estimated uncertainty in the background is one standard deviation or 0.02 microamps. This is based on the results of intercomparisons for Eras 1, 2, and 3 or for Eras 4 and 5 by the statistics from the three long standing NOAA sites as discussed in section 3.1. If a background was outside of the defined limits for its era and was reduced to the mean, the uncertainty in the background was taken as two standard deviations or 0.04 microamps.

### 4.2 Uncertainty of Flowrate

To estimate the uncertainty in the volumetric flow rate of the pump, the uncertainty in the measurement of the flowrate using a 100 cc soap bubble flowmeter, the uncertainty in the flowrate correction for the surface measurement, and the uncertainty in the pump efficiencies at low pressures are added in quadrature and is represented by Eqn. 17:

$$\left(\frac{\Delta\Phi_P}{\Phi_P}\right)^2 = \left(\frac{\Delta\Phi_{P,Meas}}{\Phi_{P,Meas}}\right)^2 + \left(\frac{\Delta C_{PF,SM}}{C_{PF,SM}}\right)^2 + \left(\frac{\Delta\eta_{PF,LP}}{\eta_{PF,LP}}\right)^2$$

$$(17)$$

The uncertainty in taking the flow rate measurement with the stop watch and bubble flow meter, $\Delta\Phi_{P,Meas}$, is estimated to be 0.5% or approximately ±0.15 seconds. The uncertainty in the flowrate correction for the surface measurement, $\Delta C_{PF,SM}$, is different for the climatological flow rate corrections and the flow rate corrections measured on the day of the flight. The uncertainty in the climatology was estimated by finding the largest standard deviation for the ambient pressure, temperature, and humidity for a given site and given month. The greatest standard deviation in ambient pressure was found to be ±15 mb at Summit Station. The greatest standard deviation in temperature and relative humidity occurred in American Samoa and was ±2.5 degrees K and ±15% respectively. The uncertainty in the pump temperature/ambient temperature difference for the climatology is estimated as ±1.5 degrees K. The new NOAA SOP for calculating the flowrate correction is to use an iMet to





find the ambient temperature, pressure and relative humidity and use the actual pump temperature during the flow rate measurement. The uncertainty in the iMet measurements is provided by the manufacturer; the pressure is ±0.5 mb, the temperature is ±0.2 degrees K and the humidity is ±5%. The uncertainty in the pump temperature/ambient temperature difference for the measured flow rate correction is estimated as ±1 degree K. With a range of conditions, the highest possible

and lowest possible flowrate correction was calculated for the climatological and measured flowrate corrections. The uncertainty of each type of correction was estimated to be half of the range of the high and low corrections. The uncertainty in the climatological flowrate corrections for the surface measurement was estimated as ±1.25% and the uncertainty of the measured flowrate correction for the surface measurement was estimated as 0.5%.

The relative uncertainty of the pump efficiency is taken as the one standard deviation of the pump efficiency average. Older

rectangular cross-section Teflon pumps used in earlier ECC ozonesonde models (1A and 3A) have not had the pump efficiency measured using the techniques in Johnson et al. (2002). As discussed earlier in Section 3.2.2, measurements of the 3A pump efficiency (Komhyr and Harris, 1971) using a bag inflation method determined 3A pump efficiencies not to dissimilar to those measured for the cylindrical cross-section pumps by Johnson et al., (2002). Measurements of the pump efficiency using the same technique for both pump configurations found that the rectangular cross-section pumps were less efficient than the

cylindrical cross-section pumps (Torres, 1981). Taking this into account, the uncertainty for the pump flowrate efficiency at low pressures for 1A and 3A ozonesondes were doubled to account for this difference.

**4.3 Uncertainty of Pump Temperature**

The uncertainty of the temperature of the pump is estimated by adding in quadrature the uncertainty of the thermistor and the electronics measuring the temperature, the uncertainty of the pump temperature difference to the temperature of the base of

the pump, and the uncertainty of the correction for the temperature of the base of the pump to the internal piston temperature and is represented by Eqn. 18:

$$\left(\frac{\Delta T_P}{T_P}\right)^2 = \left(\frac{\Delta T_{P,Meas}}{T_{P,Meas}}\right)^2 + \left(\frac{\delta(\Delta T_{P,CIB})}{T_{P,Meas}}\right)^2 + \left(\frac{\delta(\Delta T_{P,CIP})}{T_{P,Meas}}\right)^2 \tag{18}$$

The uncertainty of the measurement of the pump temperature, $\Delta T_{P,Meas}$, is estimated to be 1 degree K for the analog soundings and 0.5 degrees K for the digital sounding systems. (Smit and the O3S-DQA-Panel, 2012) The uncertainty for correcting the

1A, 3A, and 4A ozonesondes to the pump temperature measured inside the pump block, $\delta(\Delta T_{P,CIB})$, accomplished by Eqns. 8, 9 and 10 is 1.0 K. The uncertainty associated with correcting 5A ozonesondes to the pump temperature measured inside the pump block, $\delta(\Delta T_{P,CIB})$, accomplished by Eqns. 11 and 12 is 0.5 K. The uncertainty for correcting all ozonesondes from the temperature inside the pump block to the internal piston temperature, $\delta(\Delta T_{P,CIP})$, in Eqn. 13 is also 0.5K.

**4.4 Uncertainty of Ozone Sensor Efficiency**

The uncertainty in the ozone sensor efficiency is obtained by adding in quadrature the uncertainty in the absorption efficiency and the uncertainty in the conversion efficiency Eqn. 19:



$$\left(\frac{\Delta\eta_{OS}}{\eta_{OS}}\right)^2 = \left(\frac{\Delta\eta_A}{\eta_A}\right)^2 + \left(\frac{\Delta\eta_C}{\eta_C}\right)^2 \qquad (19)$$

The absorption efficiency is assumed to be 1 with an estimated uncertainty of ±1%. (Davies et al., 2003) The conversion efficiency is assumed to be 1 after the ozone sensor efficiency has been applied. It was assumed that the conversion efficiency was dominated by the stoichiometry; the stoichiometry of the reaction is estimated to have an uncertainty of ±3% (Dietz et al.,

1973). The estimate for the stoichiometry by Dietz et al. (1973) was for a buffered potassium iodide solution. Due to the 2% KI, No Buffer Solution not being buffered, the conversion efficiency uncertainty was increased to ±4.5%.

## 5 Results of Uncertainty Determination

Quantifying the uncertainty of each variable used in the ozonesonde equation (Eqn. 2) on a point by point basis was one of the key goals of the homogenization project. Figure 6 shows the uncertainties of each variable as well as the total uncertainty for

an example ozone profile from Boulder, CO. The relative uncertainties of each variable in Figure 6 are added in quadrature to obtain the total uncertainty as shown in Eqn. 16. Every profile in the NOAA long-term ozonesonde record now has a unique uncertainty estimate similar to this.

The relative uncertainty of the measured cell current and background current are the largest contributor to the overall uncertainty in the troposphere, when the difference in the measured and background cell current is the smallest. When the cell

current reaches its minimum at the tropopause at approximately 9.5 km in Figure 6, the uncertainty of the measured/background cell current reaches its maximum of approximately 6.5%. As the ozonesonde measures higher amounts of ozone through the ozone peak from 10-25 km, the difference in the measured and background cell currents becomes larger, making the uncertainty smaller. This is of greater importance at tropical sites where very low ozone values are observed through the troposphere. The measured/background uncertainty is the main contributor to the differences in the average uncertainty

observed in the troposphere for Eras 1 and 2 compared to Eras 3, 4, and 5 in Figure 7. This is because more backgrounds were reduced in Eras 1 and 2 causing a larger uncertainty in the background current and thus a larger average uncertainty. The measured/background uncertainty also plays a large role in the average uncertainty plot for the month of October for South Pole station in Figure 7. The Antarctic ozone hole forms in September and October and is easily recognizable in the average ozone partial pressure with low ozone values observed from 12-22 km. However, in this case Era 1 and 2 show lower average

uncertainties, contrary to the average uncertainties in the troposphere. This is because in Eras 1 and 2 (1967-1982 and 1982-1991, respectively) the ozone hole was not as severe as in the later eras; the ozone partial pressure did not get as low through the ozone peak and therefore the difference in the measured and background cell current did not become very small leading to a lower average uncertainty in this region for those earlier eras.

The relative uncertainty of the volumetric pump flowrate in Figure 6 is 1.6% at the surface and increases with altitude to 2.3%

at 30 km. This increase with altitude is due to the uncertainty of the ozonesonde pump efficiency loss at low pressures. It should be noted that the NOAA ozonesonde records use an average pump efficiency and the uncertainty is taken as one





standard deviation of many pump efficiency measurements. If the pump efficiency was measured and a unique pump efficiency is used to calculate ozone partial pressure, the uncertainty would be the uncertainty of the pump efficiency measurement. With an accurate and repeatable pump efficiency measurement for individual pumps, the uncertainty in the pump flowrate and thus the total uncertainty can be reduced. Figure 6 is using a climatological pump flowrate correction for the surface measurement.

When the pump flowrate correction for the surface measurement is measured during the flight preparation, the uncertainty of the pump flowrate at the surface is reduced to approximately 1.1%.

The pump temperature uncertainty is the smallest contributor to the total uncertainty through the entire record. While the pump temperature uncertainty appears to be constant, it is changing as the pump temperature changes through the flight. The earlier ozonesonde types, 1A, 3A, 4A, and 5A, have a larger uncertainty than the example profile in Figure 6 because of the added

step to homogenize the pump temperature measurement to the inside of the pump block. For 1A, 3A and 4A ozonesondes correcting to the pump temperature inside the pump block with Eqns. 8, 9, and 10 adds approximately 0.33% to the pump temperature uncertainty for a pump temperature of 300 K and for 5A ozonesondes the correction to the inside of the pump block with Eqns. 11 and 12 adds 0.17%. The added uncertainty for correcting the pump temperature from inside the pump block to the internal piston temperature which is applied to all ozonesonde types by Eqn. 12 is also 0.17% for a pump

temperature of 300 K. This results in the pump temperature uncertainty being largest for Era 1, 2 and 3.

The uncertainty of the ozone sensor efficiency is consistent for each site and is the same for all eras except Era 4 which was increased due to the unbuffered solution. This difference can be seen in Figures 7 and 8 where the average uncertainty for Era 4 is larger than Era 3, except in cases where the measured/background uncertainty is dominating the total uncertainty. The ozone sensor efficiency uncertainty is a large contributor to the total uncertainty throughout the profile. Further testing and

comparisons at the WCCOS will lead to a better understanding of the ozone sensor efficiency and possibly a reduction in its uncertainty.

For a majority of profiles at the various sites and through the various eras, the total uncertainty in the troposphere is dominated by the measured/background cell current and the ozone sensor efficiency uncertainties. In the stratosphere the largest contributors to the total uncertainty are the ozone sensor efficiency and pump flowrate uncertainty.

The total uncertainty has improved over time as the uncertainty is lower for each subsequent era except for Era 4 in some cases as shown in Figure 7 and 8. To illustrate the uncertainty range from surface to balloon burst, the total column ozone is also given with the uncertainty in Dobson Units. This is calculated by multiplying the average relative uncertainty of Eras 2, 3, 4 and 5 to the average ozone partial pressure to obtain the average absolute uncertainty. The average absolute uncertainty is then added to and subtracted from the average ozone partial pressure. The total column ozone is then calculated for the high and

low ozone partial pressure profiles and the total column uncertainty is simply half of this range. The average relative total column uncertainty for April in Dobson units as shown in Figures 7 and 8 are 4.4%, 4.2%, 4.1%, and 4.2% for Boulder, Hilo, Samoa, and South Pole respectively.



## 6 Comparisons with Satellite Total Column and Profile Measurements and Ground-based Total Column Ozone

To gauge the efficacy of the ozonesonde homogenization, the total column ozone values calculated from the ozonesonde were compared to Dobson spectrophotometers. To calculate the residual total column ozone above balloon burst, the SBUV add-on tables produced by McPeters et al. (2013) were used. If the balloon burst at a pressure greater than 7mb, the residual column ozone was calculated from 7mb. The Dobson instruments at Boulder, South Pole, Hilo, and Samoa are collocated (within 30 km) with the ozonesonde launch site and taken on the same day as the ozonesonde profile measurement. (Evans et al., 1017) Figures 9, 10, 11, and 12 show the percent difference (Ozonesonde – Dobson) in the total column ozone of the two instruments. 76.4%, 66.7%, 77.7%, and 71.2% and of the total column ozone comparisons between the ozonesonde and the Dobson are ± 5% and 95.7%, 92.2%, 96.6%, and 94.1 are ±10% for Boulder, South Pole, Hilo, and Samoa respectively. The averages of the comparisons were 0.7%, 1.98%, -0.07%, and 0.1%, and the standard deviations for the comparisons were 4.8%, 5.5%, 4.6%, and 5.0%. The higher average at the South Pole can be attributed to the need for a smaller ozone cross section in Dobson processing needed at the South Pole for the very low temperatures; the larger standard deviation is due to the difficulty of making accurate Dobson comparisons due to the low zenith angle. The Dobson uncertainty is not taken into account in this comparison, but is considered to be ± 3%. (Basher, 1985)

To gain further knowledge of the accuracy of the shape of the ozone profile, the ozonesonde data were compared to the SBUV satellite record. The SBUV satellite record of both total column and stratospheric profile measurements covers the major portion of the ozonesonde record reprocessed in this work beginning in 1970. The merged SBUV version 8.6 column ozone record has been shown to have a consistent time series with offsets not exceeding ±3% while layer average offsets fall within the range of 5-7% (Deland et al., 2012; McPeters et al., 2013; Frith et al., 2014). For comparison of the ozonesonde integrated column with the satellite column ozone matching criteria were ±12 hours and within a 200 km radius of the ozonesonde site location. Comparisons were carried out for SBUV Layers 1-8 (surface-24.5 hPa), Layer 9 (24.5-16.1 hPa), and Layer 10 (16.1-10.1 hPa) and are shown in Figures 13, 14, and 15. The SBUV total column comparisons were included in the Supplemental Material for all eight sites (Figures S1-S8) and matches closely with the comparison between the collocated Dobson total ozone data and the reprocessed data at the four long-term ozonesonde locations (Figures 9-12). In the SBUV column layer comparisons, results prior to applying the ozone sensor efficiency are shown in the left panel and fully reprocessed data are shown in the right panel. Corrections applicable to all eras that can be quantified independently, including background current threshold, pump temperature correction, and corrections for the pump flow rate measurement, were applied to the data in both the left and right panels. The eras encompassed by different sensing solution recipes are separated by dashed vertical lines. Various instrument versions are color coded. Data prior to mid-1990 used the analog data recording system and VIZ radiosonde. Various digital radiosondes were used after mid-1990 as explained in Section 1.3.

At Boulder (Figures 13), Hilo (Figure 14), and Samoa (Figure 15) with data prior to 1997 the largest change between the reprocessed and uncorrected data results from the correction for the sensing solution buffer (1% KI, 1.0 x buffer). This is most prominent in Layers 9 & 10 where the impact of the secondary reaction from the buffer is most prominent. Another important





change is seen at stations where the model 6A ozonesonde (designated by red dots) was used in combination with the no-buffer solution. This is most noticeable at Samoa (Figure 15) and Fiji (Figure S1) where this combination was used from 1997 to 2006. Data beginning in 1990 at all locations show deviations for column ozone falling within approximately ±5 % with an additional 1-2 % deviation prior to 1990. After 2015 at Hilo (Figures 9 and S2) there is an unexplained dip in the ozonesonde

column amount that is most strongly seen in the top layer (Layer 10 – Figure S1). Comparisons for individual layers show larger deviations for individual soundings but show that overall the reprocessing produces improved consistency of the vertical profile time series over the observational record. At Boulder in Layers 9 and 10 (Figures 8 & 9) prior to 1990, the period that encompasses analog data recording and the use of 1% KI, 1.0x Buffer Solution, the reprocessed data are on average a few percent lower than over the remainder of the record. This pattern is not readily discernable at Hilo or Samoa during this period,

however. This may reflect the limitations in the correction for the impact of the buffer secondary reaction that may not fully account for the differences in the distribution of ozone through the vertical profile.

**7 Discussion and Conclusions**

The homogenization process while long and laborious has improved NOAA's ECC ozonesonde data record in multiple ways. Having all data files in a common file format with all meta data accurately represented and creating the SkySonde Software

Package has made the data record more manageable by allowing for fast batch reprocessing of all ozonesonde files. If and when a better understanding of the less well quantified variables is realized, NOAA will be well prepared to implement the improved processing techniques. The enhanced plotting capabilities have improved the understanding of the fine details and issues seen in ozonesonde profile measurements, allowing for efficient screening of individual profiles. The reprocessing and homogenization of NOAA's long term vertical ozone profile record measured by the ECC ozonesondes has greatly improved

the agreement of the different ozonesonde types and the different sensing solution types for the five eras shown in this work. The comparison of the ozonesonde data record with the SBUV satellite data record improved in both the total column and pressure layer comparisons. For the first time, a bottom up, unique, line by line uncertainty calculation that accounts for all variables and used in calculating ozone partial pressure has been added to every flight. It is encouraging that the independently calculated uncertainty in total column (4.4%, 4.2%, 4.1%, and 4.2% - from Section 5) is very similar to the standard deviation

of the comparison with the Dobson (4.8%, 5.5%, 4.6%, and 5.0% - from Section 6). These uncertainties agree with the total column uncertainties determined for the entire reprocessed SHADOZ dataset, that includes our three tropical stations plus 11 additional sites (Thompson et al., 2017; Witte et al., 2017b). Although the uncertainty does not fully capture the Dobson comparison standard deviation, it should be noted that no filtering (besides in the cases of known instrument failures) was conducted on the NOAA ozonesonde record. This allowed for an unbiased look at the processing of the ozonesonde data. The

NOAA ozonesonde group is working on developing a screening method that would exclude ozonesonde measurements that don't meet a specific criteria. This will greatly improve the deviation observed in the comparisons. It should also be noted that



corrections in this work were not based on comparisons to other long term ozone data records. This ensures that the ozonesonde data record is independent and non-circular.

This information should make a more robust trend analysis possible narrowing the uncertainties in estimates of long term changes. There are still questions to be answered, however. The ozonesonde community would benefit from additional published pump efficiency measurements for all ozonesonde types, a deeper look into the cause of the background current, and a continued consistent comparison of ozonesonde type biases. A JOSIE campaign at the WCCOS is taking place in October and November 2017. JOSIE-2017 will focus on comparing ozonesonde profiles with the standard reference UV photometer under several types of tropical profile simulations. This will improve the understanding of the ozonesonde's ability to measure the very low ozone values found in the tropical troposphere and the impact of background cell current.

**Data Availability**

Ozonesonde Data – ftp://aftp.cmdl.noaa.gov/data/ozwv/Ozonesonde/

Dobson Spectrophotometer Data - ftp://aftp.cmdl.noaa.gov/data/ozwv/Dobson/WinDobson/

Solar Backscatter Ultraviolet Instruments Data – ftp://toms.gsfc.nasa.gov/pub/sbuv/MERGED/

**Acknowledgments**

The authors of this work would like to acknowledge Dr. Robert Evans, Glen McConville, Dorothy Quincy, Dr. Miyagawa Koji, Audra McClure, Dr. Irina Petropavlovskikh and the entire Dobson group at the NOAA Global Monitoring group for their work on the Dobson data that was used to compare ozonesonde and Dobson total column ozone amounts; also, Dr. Richard McPeters, Dr. Gordon Labow and Dr. Stacey Frith at the Goddard Atmospheric Chemistry and Dynamics Laboratory for their work on the SBUV Overpass Data set that was used to compare ozonesonde and satellite vertical ozone profile measurements.



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



## Figures

**Table 1: Ozonesonde manufacturer, model, years manufactured and design changes (Half Panel)**

| Manufacturer | Model # | Years Manufactured | Ozonesonde Design and Changes |
|---|---|---|---|
| Science Pump | 1A | 1967 | Rectangular Pump/Square Teflon Sensor Cell/Rod Thermistor at Base of Ozonesonde Body/Analog Data Acquisition |
| Science Pump | 3A | 1968-1981 | Commutator Moved to Electronics Board |
| Science Pump | 4A | 1978-1995 | Cylindrical Piston Pump |
| Science Pump | 5A | 1990-1997 | Digital Data Acquisition/ Pump Temperature Thermistor Epoxied to Corner of Pump Block |
| Science Pump | 6A | 1995-present | Pump Temperature Thermistor Inside Pump Block |
| EN-SCI | 1Z | 1993 – 1998 | Different Manufacturer/Same Design as 6A |
| EN-SCI | 2Z | 1997 –present | Circular Molded Plastic Sensor Cell |

**Table 2: Amount of each chemical in grams/liter deionized water used in the five commonly used cathode sensing solutions.** (1) [Komhyrand Harris, 1971], (2) [EN-SCI 1Z Manual, 1994], (3) EN-SCI MODEL 1Z & 2Z ECC-O3_SONDES [Revised May, 1996], (4) [NOAA, 1997], (5) [NOAA, 2005] (Half Panel)

| Cathode Sensing Solution | (Grams per Liter Deionized Water) | | | |
|---|---|---|---|---|
| Solution Name | KI | KBr | Na2HPO4·12H2O | NaH2PO4·H2O |
| (1) 1.5% KI, 1.5x Buffer | 15.0 | 37.5 | 7.5 | 1.880 |
| (2) 1% KI, 1.0x Buffer | 10.0 | 25.0 | 5.0 | 1.250 |
| (3) 0.5% KI, 0.5x Buffer | 5.0 | 12.5 | 2.5 | 0.625 |
| (4) 2% KI, No Buffer | 20.0 | 0.0 | 0.0 | 0.000 |
| (5) 1.0% KI,  0.1x Buffer | 10.0 | 25.0 | 0.5 | 0.125 |





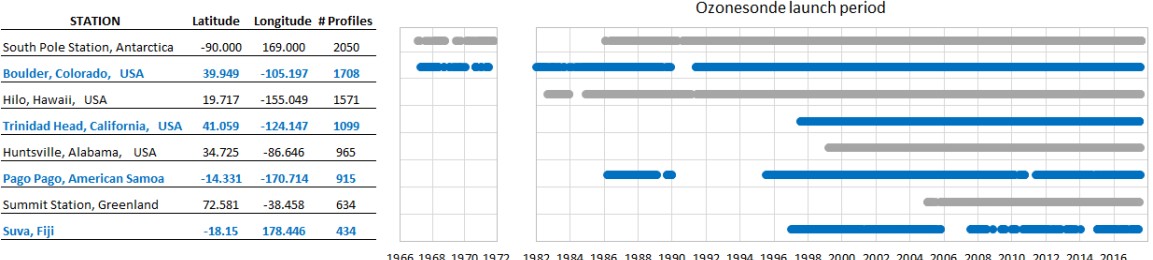

**Figure 1: The eight long-term NOAA ozonesonde stations with Latitude, Longitude, # of Profiles, and launch period. (Full Panel)**

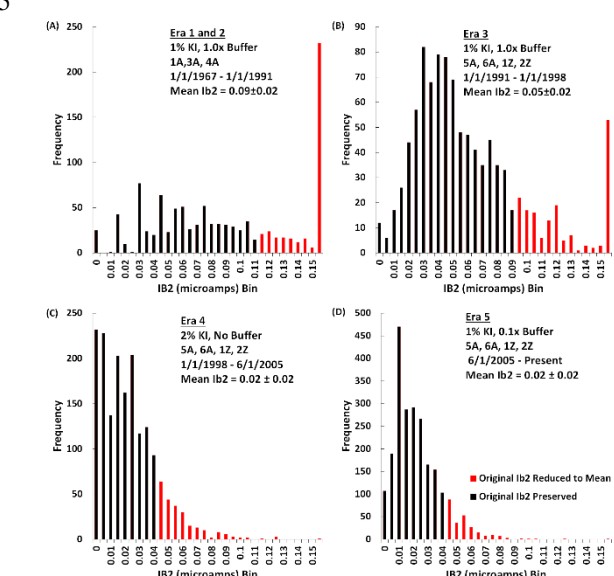

**Figure 2: Histogram of cell current backgrounds with five eras broken into four. A) Eras 1 and 2 B) Era 3 C) Era 4 D) Era 5 (Half Panel)**





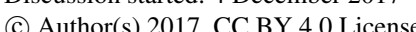

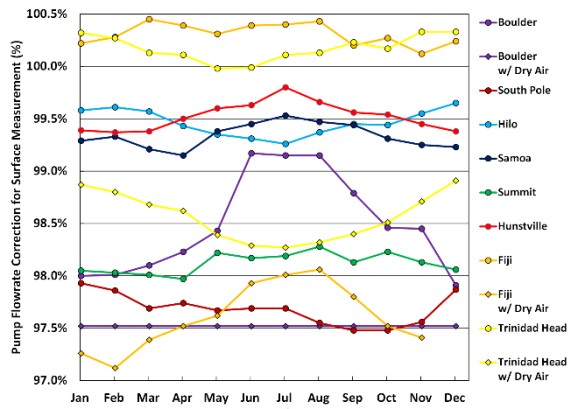

**Figure 3: Monthly Climatological Volumetric Pump Flowrate Corrections for Surface Measurement. (Half Panel)**

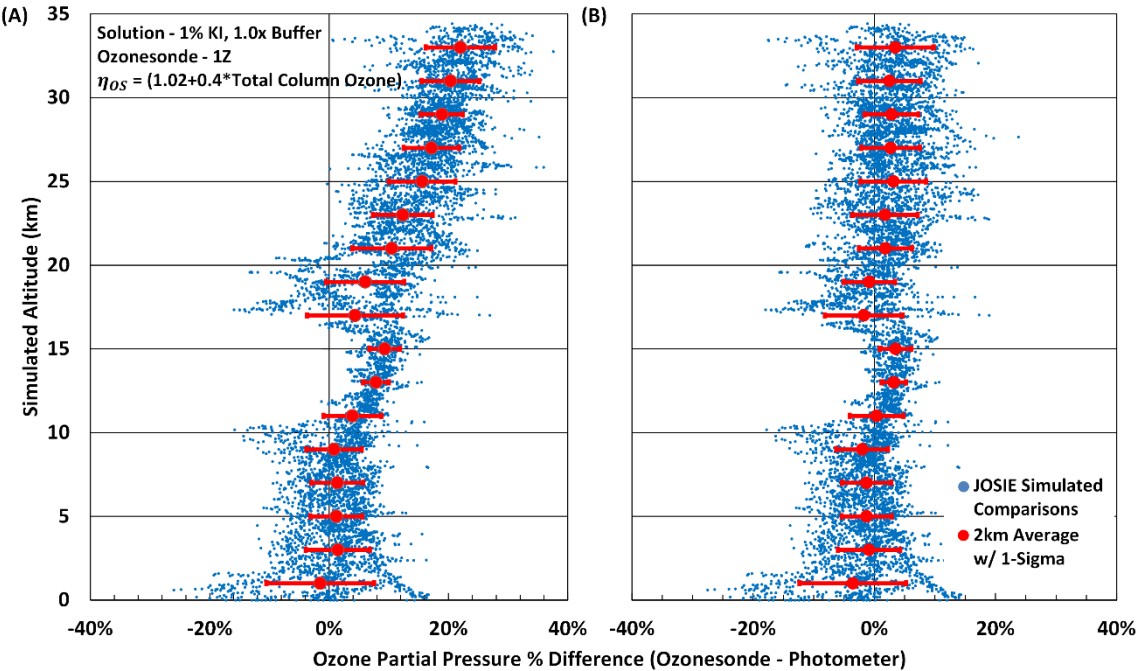

**Figure 4: Percent difference in ozone partial pressure between the ozonesonde and the reference ozone photometer with 1% KI, 1.0x**
5 **Buffer and 1Z ozonesondes before (Panel A) and after (Panel B) applying the ozone sensor efficiency. (Full Panel)**





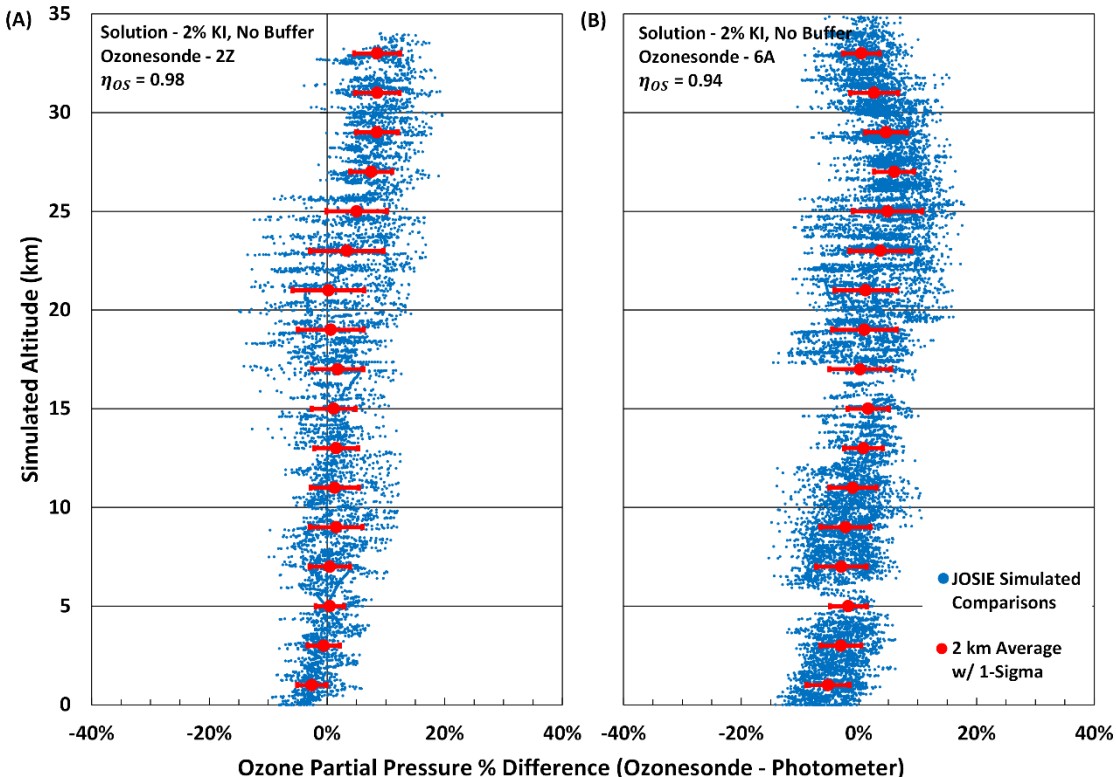

**Figure 5: Percent difference in ozone partial pressure between the ozonesonde and the reference ozone photometer with 2% No Buffer Solution and 2Z EN-SCI (Panel A) and 6A Science Pump (Panel B) ozonesondes after applying the ozone sensor efficiency. (Full Panel)**

5    **Table 3: Piece-wise functions for the uncertainty in the measured cell current of each interface board type. (Single Panel)**

| Interface Board Type | <1 μA | >1 μA |
|---|---|---|
| Analog | 0.03 | 3.0% |
| Tmax | 0.003 | 0.8% |
| V2 | 0.003 | 1.0% |
| V7 | 0.016 | 0.7% |
| X1 | 0.002 | 0.4% |





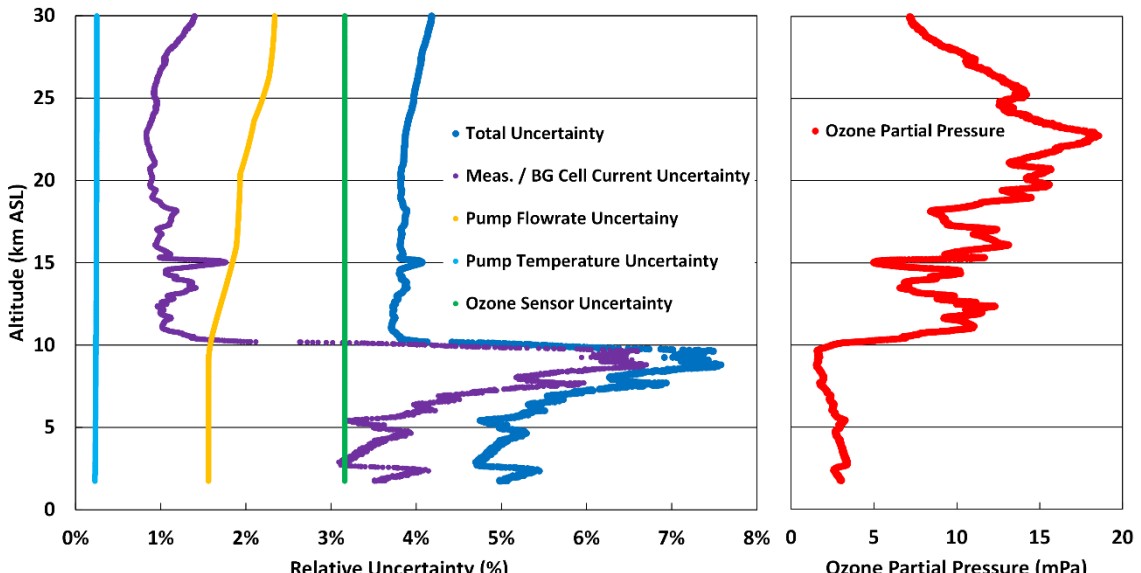

**Figure 6: Ozone partial pressure and the relative uncertainty with the relative uncertainty of each variable vs altitude for an ozone sounding in Boulder, CO. (Full Panel)**

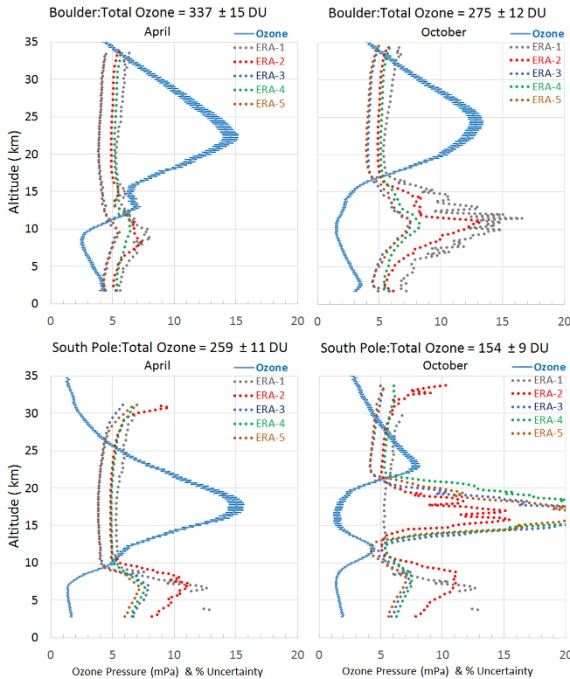

5    **Figure 7: Ozone partial pressure and average relative uncertainty for each era vs altitude for Boulder, CO and South Pole for April and October. (Half Panel)**



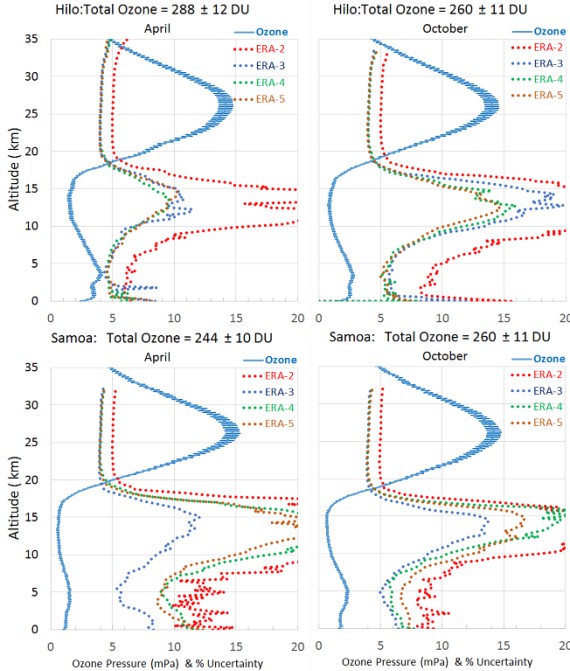

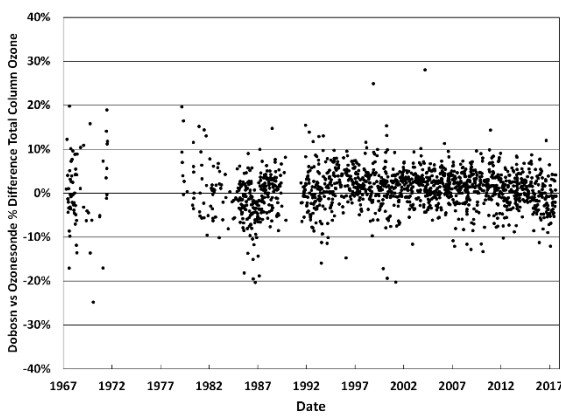

**Figure 8: Ozone partial pressure and average relative uncertainty for each era vs altitude for Hilo, HI and Pago Pago, American Samoa for April and October. (Half Panel)**

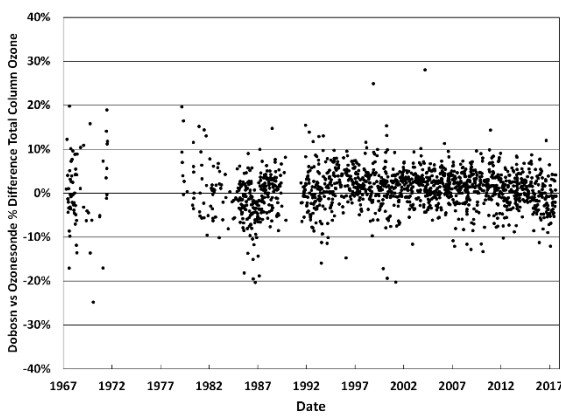

5    **Figure 9: Boulder Dobson vs Ozonesonde total column ozone comparison.**





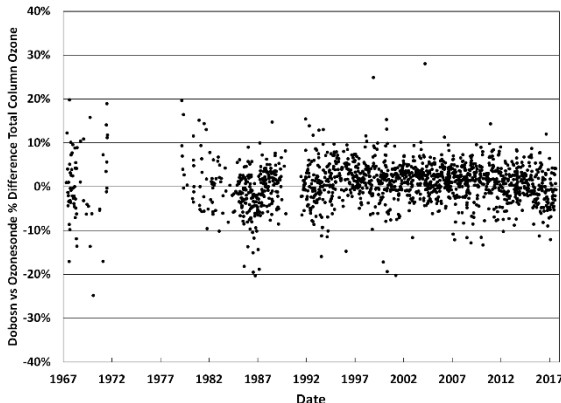

**Figure 10: South Pole Dobson vs Ozonesonde total column ozone comparison.**

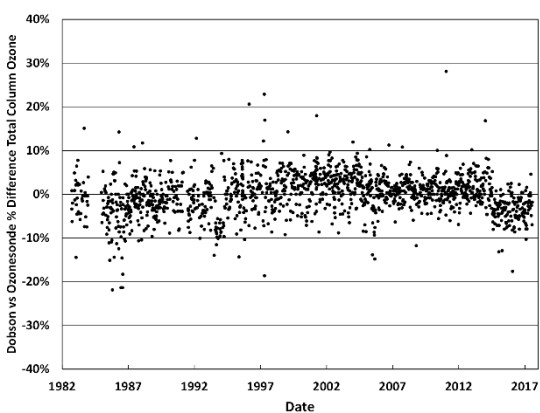

**Figure 11: Hilo Dobson vs Ozonesonde total column ozone comparison.**

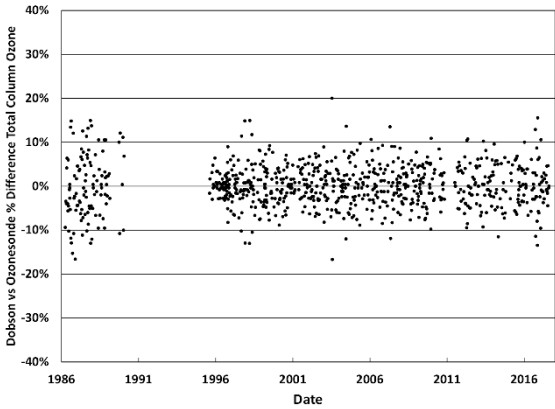

**Figure 12: Samoa Dobson vs Ozonesonde total column ozone comparison.**





**Figure 13: Percent difference in column ozone between the merged SBUV ozone data and the ozonesonde data at Boulder, CO for Layers 1-8 (Surface - 25.45 hPa), Layer 9 (25.45 - 16.06 hPa), and Layer 10 (10.13 - 16.06 hPa). Panels A, C, and E show before and Panels B, D and F show after applying the ozone sensor efficiency. (Half Panel)**



**Figure 14: Percent difference in column ozone between the merged SBUV ozone data and the ozonesonde data at Hilo, HI for Layers 1-8 (Surface - 25.45 hPa), Layer 9 (25.45 - 16.06 hPa), and Layer 10 (10.13 - 16.06 hPa). Panels A, C, and E show before and Panels B, D and F show after applying the ozone sensor efficiency. (Half Panel)**

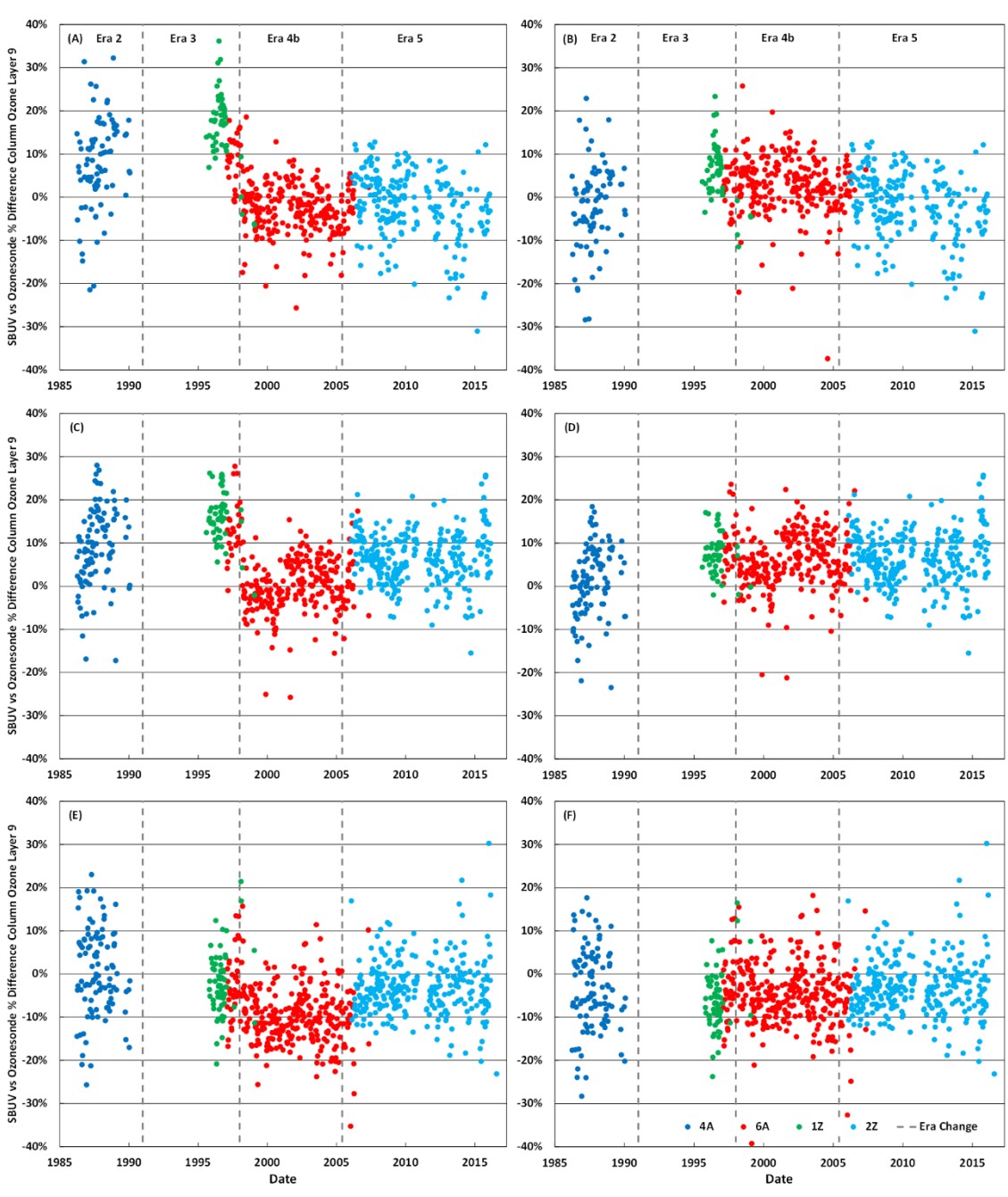

**Figure 15: Percent difference in column ozone between the merged SBUV ozone data and the ozonesonde data at Pago Pago, Samoa for Layers 1-8 (Surface - 25.45 hPa), Layer 9 (25.45 - 16.06 hPa), and Layer 10 (10.13 - 16.06 hPa). Panels A, C, and E show before and Panels B, D and F show after applying the ozone sensor efficiency. (Half Panel)**