# Peer review of "Homogenizing and Estimating the Uncertainty in NOAA's Long Term Vertical Ozone Profile Records Measured with the Electrochemical Concentration Cell Ozonesonde"

_Atmospheric Measurement Techniques, 2017_

## Referee Comment (RC1) · Anonymous Referee #1 · 22 Dec 2017

**GENERAL COMMENTS**

The paper describes the homogenization process (including uncertainty estimation) of the NOAA network of ozonesonde stations. I really enjoyed reading the very well written manuscript (although it is somewhat lengthy at some places). The authors give a nice historical overview of the ozonesonde measurements and describe in a very clear way the different instrumental effects that have to been taken into account in the homogenization process. The uncertainty analysis, developed within the O3S-DQA activity, has been applied on the profiles recorded at the NOAA network and has

been presented extensively. The impact of the homogenization of the ozonesonde data record has been assessed with Dobson total ozone data and with SBUV profile measurements. So, the research is really well established.

One of the major new achievements of the paper is the proposed approach of taking the measured (higher than the historical) pump flowrate corrections and correcting for the ozone sensor efficiency, which is derived from the comparisons of ozonesondes and the reference UV photometer at JOSIE campaigns. This is an alternative approach as the current O3S-DQA guidelines, which have set two standards (SPC 1% KI 1.0B and EN-SCI 0.5%KI 0.5B), which, with two different (historical, low) pump flowrate corrections, are within a few percent from the UV photometer at JOSIE. The authors argue that, a positive bias in the ozone sensor measurements, created by side reactions of the phosphate buffers, has to be compensated by using too low pump flowrate corrections. However, as these guidelines are still in use today, I would propose that the authors apply the O3S-DQA corrections strictly for the Eras in which one of those standards is used at NOAA sites (parts of Eras 1, 2, 3 are using the SPC 1% KI 1.0B) – so using the Komhyr (1986) pump flow correction factors – and compare those corrections with their proposed corrections (applying Eq. (15), which is based on 6 simulated flights during JOSIE 1996). By e.g. showing the average (and standard deviations) of the differences between the vertical profiles corrected by either approach, the authors should be able to demonstrate that their approach is equivalent to the O3S-DQA guidelines.

SPECIFIC COMMENTS

° Page 3: lines 22-26: Some more recent papers (as the ones mentioned) describe results from homogenized (according to the O3S-DQA guidelines) ozonesonde data: Van Malderen, R., Allaart, M. A. F., De Backer, H., Smit, H. G. J., and De Muer, D.: On instrumental errors and related correction strategies of ozonesondes: possible effect on calculated ozone trends for the nearby sites Uccle and De Bilt, Atmos. Meas. Tech., 9, 3793-3816, https://doi.org/10.5194/amt-9-3793-2016, 2016 (for Uccle and De Bilt)

and Christiansen, B., Jepsen, N., Kivi, R., Hansen, G., Larsen, N., and Korsholm, U. S.: Trends and annual cycles in soundings of Arctic tropospheric ozone, Atmos. Chem. Phys., 17, 9347-9364, https://doi.org/10.5194/acp-17-9347-2017, 2017 (for Scoresbysund, Ny Ålesund, Sodankylä, Eureka, and Ørland)

° Page 6, lines 2 and 6: please use consistently either hPa or mb through the manuscript. I would propose to use hPa.

° Page 7, line 12: "Figure 1 shows the many changes to the NOAA ozonesonde record." As a matter of fact, Figure 1 does not show all those changes, but just gives an idea of the length of and gaps in the time series of the different NOAA stations. It would be nice to have a graphical or tabular overview of those different changes, see my next point.

° Page 7, lines 15-25: the definition of the different Eras is described here, but a separate table (or graph) presenting the different characteristics of each Era is really needed, and not hidden as legends in Fig. 2 for example. This separate table (or graph) will make it also easier to follow the discussion of the homogenization, uncertainty analysis and comparison with Dobson and SBUV throughout the paper.

° Page 10, lines 28-31: Before making this statement, please refer to Figure 3 ("Figure 3 shows the different climatological flowrate corrections CPF,SM, expressed in percentages.") . To me, it seems that in Figure 3, only for Boulder , the flowrate corrections w/ Dry Air (please spell out "with") are more stable than those obtained without the Drierite air purifier/desiccant filter. But Figure 3 is very tiny, and it is hard to distinguish between the different symbols (circles and diamonds) and between some colours (Boulder/Samao & South Pole/Huntsville).

° Page 11, lines 26-27: Please mention in which range the University of Wyoming and the Japanese Meteorological Agency pump efficiency correction factors lie.

° Page 15, lines 8-10: it should be nice to have an overview here of which ozone sensor

efficiencies are used for which combinations of ozonesonde types and sensing solution strengths. I had to read this sentence several times before I understood its meaning, refering to some kind of table would help a lot, I suppose.

° Page 17, lines 24-28: where do these uncertainty estimates come from? Is there consistency with the document linked at on the NDACC web page (http://www-das.uwyo.edu/∼deshler/NDACC_O3Sondes/NDACC_O3sondes_WebPag.htm)?

° In section 5, and in figures 6, 7, 8. How do the resulting relative uncertainties compare with the relative uncertainties obtained in Van Malderen et al., 2016 & Tarasick et al., 2016, Witte et al., 2017b (???) for sites at similar latitudes? Of course, the approach in Tarasick et al., 2016 is different from the O3S-DQA uncertainty guidelines.

° Page 20, lines 4-5: I guess you mean here "If the balloon burst at a pressure smaller than 7 hPa, the residual column ozone was calculated from 7 hPa".

TECHNICAL CORRECTIONS

° Page 3, line 20: "based on the JOSIE and BESOS intercomparisons" instead of "based on the WMO and JOSIE intercomparisons"

° Page 6, line 25: "cannot be measured directly" instead of "cannot me measured directly"

° Page 18, line 9: "of the data quality assessment project" instead of "of the homogenization project"?

° Page 20, line 6: "Evans et al., 2017" instead of "Evans et al., 1017". I don't think Bob is that old.

° Page 21, line 4: "Figures 11 and S7" instead of "Figures 9 and S2"

° Page 21, line 5: "(Layer 10 – Figure 14)" instead of "(Layer 10 – Figure S1)"

° Page 21, line 7: "(Figure 13)" instead of "(Figures 8 & 9)"

○ Page 28, line 28: the Thompson et al. JGR 2017 paper is now published

○ Page 31, caption Figure 2: please add that those histograms are taken for the measurements at the sites Boulder, South Pole & Hilo

---

## Referee Comment (RC2) · Anonymous Referee #3 · 1 Jan 2018

General comments

The authors can be congratulated for this important and comprehensive study! It is a major step in the effort for a global homogenization of the ozonesonde data sets. I rate the overall quality as excellent. I recommend publication after some minor revisions.

Specific comments

1. Page 5, Line 32: "RS-80 pressure sensors are known to have degraded over time." How is this meant? Are they aging or became production worse? Please give refer-

ence.

2. P6L1-3: Is this part of this study? If yes, give more details. If not, give a reference.

3. P7L10-11: Don't understand this sentence.

4. P7L12-14: Figure 1 doesn't show any changes.

5. P14L22: How was it determined? Is it part of this study? Reference? Same questions for the values 0.98 and 0.94 at P15L2 and P15L7.

6. P15L9: Don't understand why 0.96 is used instead of 0.94.

7. P16L6: I assume "constant" is meant instead of "linear".

Technical corrections

1. P4L19: Delete empty space character in front of "Changes".

2. P4L24: Delete most empty spaces between "2" and "KI".

3. P6L5: Explain "SkySonde" here and not later (page 7).

4. P6L16+: Introduce variable symbols used in equations consistently in the text (when it is mentioned the first time). E.g. at this place: "... the ozone partial pressure, $P\_O3$, is determined ...". An introduction is missing or too late at many other places, e.g. P9L25, P10L11,12, P12L12,13. Please use a consistent notation: "..., symbol, ..." or "... (symbol) ..." but not both.

5. P6L25: "cannot BE measured"

6. P7L16: First occurrence of the notion n.nx buffer solution in the main text. Please give a hint that the notion is defined in table 2.

7. P7L16+: Write "buffer solution" in a consistent way with upper or lower characters throughout the text.

8. P10L10+: Get the subscript depths right.

9. P12,13 Equations 8-13: Add unit "K".

10. P12L21, P17L23: "degree" is not part of the unit Kelvin. Please delete.

11. P13L1: Exchange "truest" by "best estimate of the".

12. P16L14,17: Please use "ïA■A" instead of "microamps".

13. P16L20: Please use "cm**3" instead of "cc".

14. P16L22: Check place of equation number.

15. P17L26,27,28: Add "estimated" before temperature, e.g. "estimated 1.0 K".

16: P17L28: Add a space between "0.5" and "K".

17: P20L8: Delete on "and".

18: P20L9: Add "… average DIFFERENCES of the …".

19. P20L23 & Figures S5-S8: I assume the captions for S5-S8 mentioning Dobson instead of SBUV are wrong.

20. P21L4: "Figures 11 and S7"

21. P21L7: "(Figure 13)" instead of "(Figures 8 & 9)"

22. Figure 1: What is the meaning of a longitude of 169 (East of West?) at a latitude of -90. Please add East and North units.

23. F1: Good place to mark the different eras graphically.

24. F2: Explain large bars at the end of histograms (A) and (B).

25. F3: Use lower case characters.

26. Table 3: Add units for second column.

27. F9-12: It would be nice to have the eras mark as in later figures.

28. F13-15: The relation layer to panel character is somewhat hidden. Please repeat in the caption.

29. F13-15, FS1-8: Explain colour code.

---

## Author Comment (AC1) · 7 Mar 2018

The authors would like to thank the referees for the constructive criticism of our manuscript. We have outlined our responses to the reviewers' comments as well as the subsequent changes to the manuscript in the following response.

Anonymous Referee #1 GENERAL COMMENTS: The paper describes the homogenization process (including uncertainty estimation) of the NOAA network of ozonesonde stations. I really enjoyed reading the

very well written manuscript (although it is somewhat lengthy at some places). The authors give a nice historical overview of the ozonesonde measurements and describe in a very clear way the different instrumental effects that have to been taken into account in the homogenization process. The uncertainty analysis, developed within the O3SDQA activity, has been applied on the profiles recorded at the NOAA network and has been presented extensively. The impact of the homogenization of the ozonesonde data record has been assessed with Dobson total ozone data and with SBUV profile measurements. So, the research is really well established. One of the major new achievements of the paper is the proposed approach of taking the measured (higher than the historical) pump flowrate corrections and correcting for the ozone sensor efficiency, which is derived from the comparisons of ozonesondes and the reference UV photometer at JOSIE campaigns. This is an alternative approach as the current O3S-DQA guidelines, which have set two standards (SPC 1% KI 1.0B and EN-SCI 0.5%KI 0.5B), which, with two different (historical, low) pump flowrate corrections, are within a few percent from the UV photometer at JOSIE. The authors argue that, a positive bias in the ozone sensor measurements, created by side reactions of the phosphate buffers, has to be compensated by using too low pump flowrate corrections. However, as these guidelines are still in use today, I would propose that the authors apply the O3S-DQA corrections strictly for the Eras in which one of those standards is used at NOAA sites (parts of Eras 1, 2, 3 are using the SPC 1% KI 1.0B) –so using the Komhyr (1986) pump flow correction factors – and compare those corrections with their proposed corrections (applying Eq. (15), which is based on 6 simulated flights during JOSIE 1996). By e.g. showing the average (and standard deviations) of the differences between the vertical profiles corrected by either approach, the authors should be able to demonstrate that their approach is equivalent to the O3S-DQA guidelines.

Authors' Response: The reviewer makes a good point about a difference in the processing methods of NOAA and other ozonesonde sites during Eras 1, 2, and 3. NOAA desires to be consistent with its processing methods throughout the record, so chooses to keep the Johnson 2002 pump efficiencies and ozone sensor efficiency based on the

cumulative ozone exposure. To assess the difference in the processing techniques, Era 3 for Boulder, CO and Hilo, HI were processed with each processing technique and compared. Two new figures (Figures S1 and S2) have been included in the supplementary material as well as a new sentence discussing the difference in the 1986 Komhyr processing method and the NOAA Accumulating Buffer Bias Correction. The figure plots the average Boulder and Hilo ozone profiles processed with both methods and a % Difference plot for each. The difference is less than the uncertainty for these data, so we have deemed it neglible. Additionally, the 1986 Komhyr processing method would increase the partial pressure of ozone in comparison to the NOAA approach. This increase would make the comparison with the ozone photometer at JOSIE worse and the comparison with the SBUV measurements worse. What is now Page 15 Line 4 now includes, "NOAA's approach (ozone sensor efficiency) differs from the ASOPOS standard processing for SPC ozonesondes paired with 1.0% KI, 1.0X Buffer Solution (1986 Komhyr corrections) in Eras 1, 2, and 3. To compare the two processing methods, the average profiles for Boulder and Hilo for Era 3 are shown in panel A of Figures S1 and S2, respectively, processed with the NOAA approach and the ASOPOS approach. The percent difference is included on panel B of the plots and the difference is less than the uncertainty of the ozone measurement for these eras."

SPECIFIC COMMENTS âŮę Page 3: lines 22-26: Some more recent papers (as the ones mentioned) describe results from homogenized (according to the O3S-DQA guidelines) ozonesonde data: Van Malderen, R., Allaart, M. A. F., De Backer, H., Smit, H. G. J., and De Muer, D.: On instrumental errors and related correction strategies of ozonesondes: possible effect on calculated ozone trends for the nearby sites Uccle and De Bilt, Atmos. Meas. Tech., 9, 3793-3816, https://doi.org/10.5194/amt-9-3793-2016, 2016 (for Uccle and De Bilt) and Christiansen, B., Jepsen, N., Kivi, R., Hansen, G., Larsen, N., and Korsholm, U.S.: Trends and annual cycles in soundings of Arctic tropospheric ozone, Atmos. Chem. Phys., 17, 9347-9364, https://doi.org/10.5194/acp-17-9347-2017, 2017 (for Scoresbysund, Ny Ålesund, Sodankylä, Eureka, and Ørland) Authors' Response: Both citations were added to the text. âŮę Page 6, lines 2 and

6: please use consistently either hPa or mb through the manuscript. I would propose to use hPa. Authors' Response: Changed all mentions of mb to hPa. âŮę Page 7, line 12: "Figure 1 shows the many changes to the NOAA ozonesonde record." As a matter of fact, Figure 1 does not show all those changes, but just gives an idea of the length of and gaps in the time series of the different NOAA stations. It would be nice to have a graphical or tabular overview of those different changes, see my next point. Authors' Response: Figure 1 has been updated to show the eras and the changes in ozonesonde type, solution type, and radiosonde type. âŮę Page 7, lines 15-25: the definition of the different Eras is described here, but a separate table (or graph) presenting the different characteristics of each Era is really needed, and not hidden as legends in Fig. 2 for example. This separate table (or graph) will make it also easier to follow the discussion of the homogenization, uncertainty analysis and comparison with Dobson and SBUV throughout the paper. Authors' Response: Figure 1 now conveys the Eras and the different characteristics of each. âŮę Page 10, lines 28-31: Before making this statement, please refer to Figure 3 ("Figure 3 shows the different climatological flowrate corrections CPF,SM, expressed in percentages.") . To me, it seems that in Figure 3, only for Boulder, the flowrate corrections w/ Dry Air (please spell out "with") are more stable than those obtained without the Drierite air purifier/desiccant filter. But Figure 3 is very tiny, and it is hard to distinguish between the different symbols (circles and diamonds) and between some colours (Boulder/Samao & South Pole/Huntsville). Authors' Response: The statement "Figure 3 shows the different climatological flowrate corrections CPF,SM, expressed in percentages." has been added to what is now Page 11 Line 8. Figure 3 has been updated to increase the size of the symbols, the size of the graph, and to spell out with. The stability or variation and the uncertainty of the flowrate correction for the surface measurement depend on different factors. The uncertainty of the flowrate correction is based on the uncertainty of the ambient temperature and pressure of the room, the uncertainty of the humidity of the air stream being sampled, and the uncertainty of the pump/ambient temperature difference. The stability of the flowrate

correction is dependent on the climatological range of these factors. âŮę Page 11, lines 26-27: Please mention in which range the University of Wyoming and the Japanese Meteorological Agency pump efficiency correction factors lie. Authors' Response: On what is now Page 12 line 3, the sentenced has been updated to read, "These PCF's agree nicely with the Johnson et al. (2002), Wyoming (Harder, 1987) and Japan Meterological Agency's PCF's (Private communication, Tatsumi Nakano) of 1.145, 1.120, and 1.122 at 10 hPa and 1.260, 1.224, and 1.213 at 5 hPa respectively." âŮę Page 15, lines 8-10: it should be nice to have an overview here of which ozone sensor efficiencies are used for which combinations of ozonesonde types and sensing solution strengths. I had to read this sentence several times before I understood its meaning, referring to some kind of table would help a lot, I suppose. Authors' Response: A new table has been added, Table 3, which shows the ozonesonde types and sensing solutions with their corresponding ozone sensor efficiency. A new sentence was added on what is now Page 15, Line 28. "Table 3 summarizes the ozone sensor efficiencies used for all ozonesonde type and sensing solution pairings." âŮę Page 17, lines 24-28: where do these uncertainty estimates come from? Is there consistency with the document linked at on the NDACC web page (http://wwwdas.uwyo.edu/âĹijdeshler/NDACC_O3Sondes/NDACC_O3sondes_WebPag.htm)? Authors' Response: The uncertainty estimates does come from Smit, H. G. J. and the O3S-DQA-Panel (Ozone Sonde Data Quality Assessment): Guidelines for ho­mogenization of ozonesonde data, SI2N/O3S-DQA activity as part of "Past changes in the vertical distribution of ozone assessment" document on the NDACC webpage. The citation has now been included for these estimates. âŮę In section 5, and in figures 6, 7, 8. How do the resulting relative uncertainties compare with the relative uncertainties obtained in Van Malderen et al., 2016 & Tarasick et al., 2016, Witte et al., 2017b (???) for sites at similar latitudes? Of course, the approach in Tarasick et al., 2016 is different from the O3S-DQA uncertainty guidelines. Authors' Response: A new sentence has been added on what is now Page 20 Line 18. "The total relative uncertainty of ozone with altitude are similar in shape and comparable in magnitude to

other recent ozonesonde uncertainty estimates, Van Malderen et al. 2016, Tarasick et al. 2016, and Witte et al. 2017b." âŮę Page 20, lines 4-5: I guess you mean here "If the balloon burst at a pressure smaller than 7 hPa, the residual column ozone was calculated from 7 hPa". Authors' Response: Yes, that is what was meant and is corrected in manuscript.

TECHNICAL CORRECTIONS âŮę Page 3, line 20: "based on the JOSIE and BESOS intercomparisons" instead of "based on the WMO and JOSIE intercomparisons" Authors' Response: Corrected in manuscript. âŮę Page 6, line 25: "cannot be measured directly" instead of "cannot me measured directly" Authors' Response: Corrected in manuscript. âŮę Page 18, line 9: "of the data quality assessment project" instead of "of the homogenization project"? Authors' Response: Corrected in manuscript. âŮę Page 20, line 6: "Evans et al., 2017" instead of "Evans et al., 1017". I don't think Bob is that old. Authors' Response: Corrected in manuscript. âŮę Page 21, line 4: "Figures 11 and S7" instead of "Figures 9 and S2" Authors' Response: Corrected in manuscript. âŮę Page 21, line 5: "(Layer 10 – Figure 14)" instead of "(Layer 10 – Figure S1)" Authors' Response: Corrected in manuscript. âŮę Page 21, line 7: "(Figure 13)" instead of "(Figures 8 & 9)" Authors' Response: Corrected in manuscript. âŮę Page 28, line 28: the Thompson et al. JGR 2017 paper is now published Authors' Response: Corrected in manuscript. âŮę Page 31, caption Figure 2: please add that those histograms are taken for the measurements at the sites Boulder, South Pole & Hilo Authors' Response: Corrected in manuscript. Now reads, "Histogram of all cell current backgrounds from Boulder, South Pole and Hilo broken into four time periods. A) Eras 1 and 2 B) Era 3 C) Era 4 D) Era 5"

The authors would like to again thank the reviewers for doing a thorough job of reviewing the manuscript. It improved the paper a great deal. A few other grammatical and formatting errors that did not change the meaning or intention of the text were found and corrected during the process of responding to the reviews.

[Figure]

[Figure]

[Figure]

**Fig. 1.** Figure 1: The eight long-term NOAA ozonesonde stations with Latitude, Longitude, # of Profiles, and launch period.

[Figure]

[Figure]

**Fig. 2.** Figure S1: Average Boulder profile for Era 3 processed with the 1986 Komhyr processing and the NOAA ozone sensor efficiency processing techniques, Panel A. The percent difference in the two processing

**Fig. 3.** Figure S2: Average Hilo profile for Era 3 processed with the 1986 Komhyr correction and the NOAA ozone sensor efficiency correction, Panel A. The percent difference in the two processing is shown in P

---

## Author Comment (AC2) · 7 Mar 2018

Anonymous Referee #3 The authors would like to thank the referees for the constructive criticism of our manuscript. We have outlined our responses to the reviewers' comments as well as the subsequent changes to the manuscript in the following response.

General comments: The authors can be congratulated for this important and comprehensive study! It is a major step in the effort for a global homogenization of the

ozonesonde data sets. I rate the overall quality as excellent. I recommend publication after some minor revisions. Specific comments: 1. Page 5, Line 32: "RS-80 pressure sensors are known to have degraded over time." How is this meant? Are they aging or became production worse? Please give reference. Authors' Response: The sentence "RS-80 pressure sensors are known to have degraded over time." has been removed from the manuscript text. This RS-80 issue is described in more detail in the next response. 2. P6L1-3: Is this part of this study? If yes, give more details. If not, give a reference. Authors Response: When Vaisala stopped manufacturing the RS80, NOAA was able to acquire and fly over 1000 inexpensive surplus RS80 radiosondes. Some of these radiosondes pressure offsets were greater than the specified uncertainty stated on the manufacturer's datasheet. In order to determine the pressure offset we performed laboratory tests with an atmospheric chamber and a calibrated surface barometer. This is described in the text on what is now Page 6 Lines 1-5. The sentence on what is now Page 6 Line 13 has been updated and now reads, "The uncertainties of the radiosondes, while important, are beyond the scope of this analysis." 3. P7L10-11: Don't understand this sentence. Authors' Response: The sentence on Page 7 Line 14 now reads, "This is in contrast to the approach of homogenizing the record to one of the ASOPOS standard ozonesonde type/solution type/pump efficiency pairing and using transfer functions to adjust for changes in the record." 4. P7L12-14: Figure 1 doesn't show any changes. Authors' Response: Figure 1 has been updated to show the eras and the changes in ozonesonde type, solution type, data acquisition, and radiosonde type. 5. P14L22: How was it determined? Is it part of this study? Reference? Same questions for the values 0.98 and 0.94 at P15L2 and P15L7. Authors' Response: The manuscript text has been updated to include the following sentence on what is now Page 13 Line 24, "The ozone sensor efficiency is determined by iteratively minimizing the percent difference in the ozonesonde and the ozone photometer for a given ozonesonde type/sensing solution pairing. Figures 4 and 5 show these differences." 6. P15L9: Don't understand why 0.96 is used instead of 0.94. Authors' Response: The ozone sensor efficiency of 0.94 is believed to be due to the 6A ozonesonde type which

requires an ozone sensor efficiency of 0.96 and the 2.0% KI, No Buffer Solution which requires a 0.98 ozone sensor efficiency which totaled is approximately 0.94. A new table, Table 3, has been added that shows the ozonesonde types and sensing solutions with their corresponding ozone sensor efficiency. A new sentence was added on what is now Page 15, Line 28. "Table 3 summarizes the ozone sensor efficiencies used for all ozonesonde type and sensing solution pairings." 7. P16L6: I assume "constant" is meant instead of "linear". Authors' Response: Yes, constant was meant, not linear. Corrected in manuscript.

Technical corrections: 1. P4L19: Delete empty space character in front of "Changes". Authors' Response: Corrected in manuscript. 2. P4L24: Delete most empty spaces between "2" and "KI". Authors' Response: Corrected in manuscript. 3. P6L5: Explain "SkySonde" here and not later (page 7). Authors' Response: What is now Page 6, Line 7 includes the sentence, "A new data acquisition and processing software called SkySonde was developed to facilitate the implementation of the corrections associated with the data quality assessment project." And what is now Page 8, Line 4 reads, "This allows the SkySonde software to read all data files and calculate all ozone values from the raw cell current and pump temperature regardless of the data acquisition system or file format previously used." 4. P6L16+: Introduce variable symbols used in equations consistently in the text (when it is mentioned the first time). E.g. at this place: ". . . the ozone partial pressure, $P\_O3$, is determined . . .". An introduction is missing or too late at many other places, e.g. P9L25, P10L11,12, P12L12,13. Please use a consistent notation: ". . ., symbol, . . ." or ". . . (symbol) . . ." but not both. Authors' Response: The variable symbols have been added where it is first mentioned in the manuscript and all instances of ,symbol, were changed to (symbol). 5. P6L25: "cannot BE measured" Authors' Response: Corrected in manuscript. 6. P7L16: First occurrence of the notion n.nx buffer solution in the main text. Please give a hint that the notion is defined in table 2. Authors' Response: The following sentence has been added to the manuscript at what is now Page 7 Line 20, "This sensing solution nomenclature and recipes are shown in Table 2." 7. P7L16+: Write "buffer solution" in a consistent way with upper

or lower characters throughout the text. Authors' Response: All instances where the name of a particular solution is being discussed have been changed to upper case letters such as 1% KI, 1.0 x Buffer Solution. When discussing buffering agents or the secondary buffer reaction, buffer was changed to lower case characters. 8. P10L10+: Get the subscript depths right. Authors' Response: Corrected in manuscript. 9. P12,13 Equations 8-13: Add unit "K". Authors' Response: The sentence on what is now Page 12 Line 19 has been updated to read, "All temperatures used in calculating ozone are in Kelvin. The pump temperature ( $T\_P$) is calculated by adding the differences between configurations and inside of the pump block ( $\Delta T\_{(P,CIB)}$), and the difference between the inside of the pump block and the internal piston temperature ( $\Delta T\_{(P,CIP)}$) to the raw pump temperature measured ( $T\_{(P,Meas)}$) with Eqn. 7:" 10. P12L21, P17L23: "degree" is not part of the unit Kelvin. Please delete. Authors' Response: Corrected in manuscript. 11. P13L1: Exchange "truest" by "best estimate of the". Authors' Response: Corrected in manuscript. 12. P16L14,17: Please use "ï ËŻA A" instead of "microamps". Authors' Response: All instances of "microamps" has been changed to $\mu$A throughout manuscript. 13. P16L20: Please use "cm**3" instead of "cc". Authors' Response: "cc" changed to cm3 throughout manuscript. 14. P16L22: Check place of equation number. Authors' Response: Corrected in manuscript. 15. P17L26,27,28: Add "estimated" before temperature, e.g. "estimated 1.0 K". Authors' Response: Corrected in manuscript. 16: P17L28: Add a space between "0.5" and "K". Authors' Response: Corrected in manuscript. 17: P20L8: Delete on "and". Authors' Response: Corrected in manuscript. 18: P20L9: Add ". . . average DIFFERENCES of the . . ." Authors' Response: Corrected in manuscript. 19. P20L23 & Figures S5-S8: I assume the captions for S5-S8 mentioning Dobson instead of SBUV are wrong. Authors' Response: Yes, the captions should read SBUV instead of Dobson, except for Figure S6 as the South Pole does not have SBUV data. All captions except for Figure S5 are corrected in manuscript. 20. P21L4: "Figures 11 and S7" Authors' Response: Corrected in manuscript. 21. P21L7: "(Figure 13)" instead of "(Figures 8 & 9)" Authors' Response: Corrected in manuscript. 22. Figure 1: What is the meaning of a longitude

of 169 (East of West?) at a latitude of -90. Please add East and North units. Authors'
Response: Figure 1 has been updated. The latitude and longitude now include East
and North units. 23. F1: Good place to mark the different eras graphically. Authors'
Response: Figure 1 has been updated to show the eras. 24. F2: Explain large bars
at the end of histograms (A) and (B). Authors' Response: The following sentence has
been added at what is now Page 9 Line 12 to explain the large bars at the end of the
histograms (A) and (B). "In Figure 2 Panels A and B, the large number of backgrounds
greater than the scale of the histograms are attributed to erroneous measurements
attributed to the degraded ozone destruct filters." 25. F3: Use lower case characters.
Authors' Response: Figure 3 has been updated and now reads "Boulder with dry air",
"Fiji with dry air", and "Trinidad Head with dry air". 26. Table 3: Add units for second
column. Authors' Response: Table has been updated and the second column now
includes the units ($\mu$A). A new table was added, so Table 3 is now Table 4. 27. F9-12:
It would be nice to have the eras mark as in later figures.' Authors' Response: Figures
9-12 have been updated to include the era marks. 28. F13-15: The relation layer to
panel character is somewhat hidden. Please repeat in the caption. Authors' Response:
To make the relationship between the panels, layers, and processing more clear, the
captions in Figures 13-15 have been changed and now reads "Percent difference in
column ozone between the merged SBUV ozone data and the ozonesonde data at
Boulder, CO. Panels E and F show Layers 1-8 (Surface - 25.45 hPa), Panels C and D
show Layer 9 (25.45 - 16.06 hPa), and Panels A and B show Layer 10 (10.13 - 16.06
hPa). Panels A, C, and E show before and Panels B, D and F show after applying the
ozone sensor efficiency." 29. F13-15, FS1-8: Explain colour code. Authors' Response:
The different colors in the plot represent different ozonesonde types. This is shown in
the legend and is consistent with all of the comparison plots.

The authors would again like to thank the reviewers for doing a thorough job of review-
ing the manuscript. It improved the paper a great deal. A few other grammatical and
formatting errors that did not change the meaning or intention of the text were found
and corrected during the process of responding to the reviews.

[Figure]

[Figure]

**Fig. 1.** Figure 1: The eight long-term NOAA ozonesonde stations with Latitude, Longitude, # of Profiles, and launch period.

[Figure]

**Fig. 2.** Figure S1: Average Boulder profile for Era 3 processed with the 1986 Komhyr processing and the NOAA ozone sensor efficiency processing techniques, Panel A. The percent difference in the two processing

[Figure]

**Fig. 3.** Figure S2: Average Hilo profile for Era 3 processed with the 1986 Komhyr correction and the NOAA ozone sensor efficiency correction, Panel A. The percent difference in the two processing is shown in P